# Learnability-Informed Fine-Tuning of Diffusion Language Models

Shubham Parashar[1]   Atharv Chagi[1]   Jacob Helwig[1]   Lakshmi Jotsna[1]   Sushil Vemuri[2]   James Caverlee[1]
Dileep Kalathil[1 2]   Shuiwang Ji[1]

## Abstract

We aim to improve the reasoning capabilities of diffusion language models (DLMs). While SFT is a popular post-training recipe for autoregressive models, its use in DLMs faces challenges and can even hurt performance, though the underlying causes remain understudied. Our analysis reveals that vanilla SFT overlooks *learnability*, namely *what* and *when* tokens are learned. Specifically, rare tokens are difficult to learn when most of the input is masked, whereas it is straightforward and thus of little value to learn common tokens when most of the input is unmasked. Motivated by our analysis, we propose **LIFT**, an efficient SFT-based post-training algorithm for DLMs. LIFT learns easy tokens when most of the input is masked and hard tokens when more context is available, thereby aligning training with the information available at different diffusion time steps. Our results show that LIFT outperforms existing SFT baselines across six reasoning benchmarks, achieving up to a $3\times$ relative gain on AIME'24 and AIME'25. Our code is publicly available at https://github.com/divelab/LIFT.

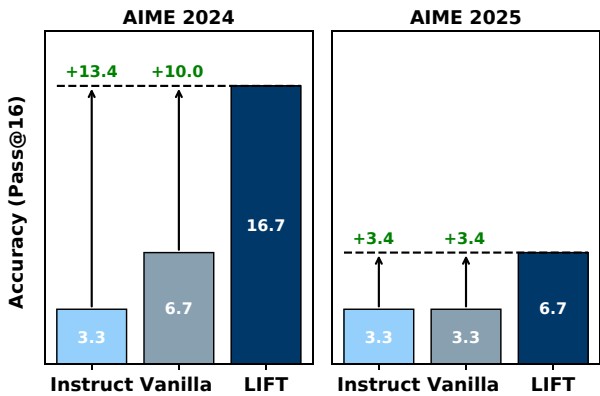

*Figure 1.* **Performance on AIME benchmarks**. Pass@16 accuracy comparison on AIME'24 and AIME'25 for LLaDA-8B-Instruct, vanilla SFT, and LIFT. LIFT achieves substantial relative improvements over vanilla SFT on both challenging mathematical reasoning datasets, demonstrating the effectiveness of learnability-informed training.

## 1. Introduction

Diffusion models have shown impressive performance in image (Song & Ermon, 2019; Nichol & Dhariwal, 2021) video (Ho et al., 2022) generation applications. Recently, diffusion models have been successfully applied to textual data, leading to the recent surge of interest in Diffusion Language Models (DLMs) (Austin et al., 2021a; Sahoo et al., 2024). A central promise of DLMs over autoregressive language models (ARLMs) is their ability to generate multiple tokens in parallel per model call, yielding substantial gains in inference throughput (Khanna et al., 2025; Wu et al., 2026). Several open-weight DLMs, such as LLaDA (Nie et al., 2025) and Dream (Ye et al., 2025), are now available, and they largely match the performance of similarly-sized ARLM counterparts.

Following the success of post-training of ARLMs to improve reasoning, recent works have explored post-training of DLMs using supervised or instruction finetuning (SFT) (Ye et al., 2025; Nie et al., 2025) and reinforcement learning (RL) (Zhao et al., 2025). However, in contrast to ARLMs, RL in DLMs is substantially more challenging both technically and algorithmically due to intractable sequence-level likelihoods, and most works on RL for DLMs propose approximations to overcome this challenge (Zhao et al., 2025; Kunde et al., 2026; Wang et al., 2025). SFT has been studied less thoroughly, and to date no work has systematically examined the challenges involved in applying SFT to DLMs. Recent results suggest that SFT can in fact degrade model performance relative to pretraining (Ye et al., 2025). This motivates the central question of our work, which we decompose into two sub-questions: (i) *what are the major factors that influence SFT post-training of DLMs*, and (ii)

---

[1]Department of Computer Science and Engineering, Texas A&M University, College Station, TX, USA [2]Department of Electrical and Computer Engineering, Texas A&M University, College Station, TX, USA. Correspondence to: Shubham Parashar <shubhamprshr@tamu.edu>, Dileep Kalathil <dileep.kalathil@tamu.edu>, Shuiwang Ji <sji@tamu.edu>.

*Proceedings of the $43^{rd}$ International Conference on Machine Learning*, Seoul, South Korea. PMLR 306, 2026. Copyright 2026 by the author(s).

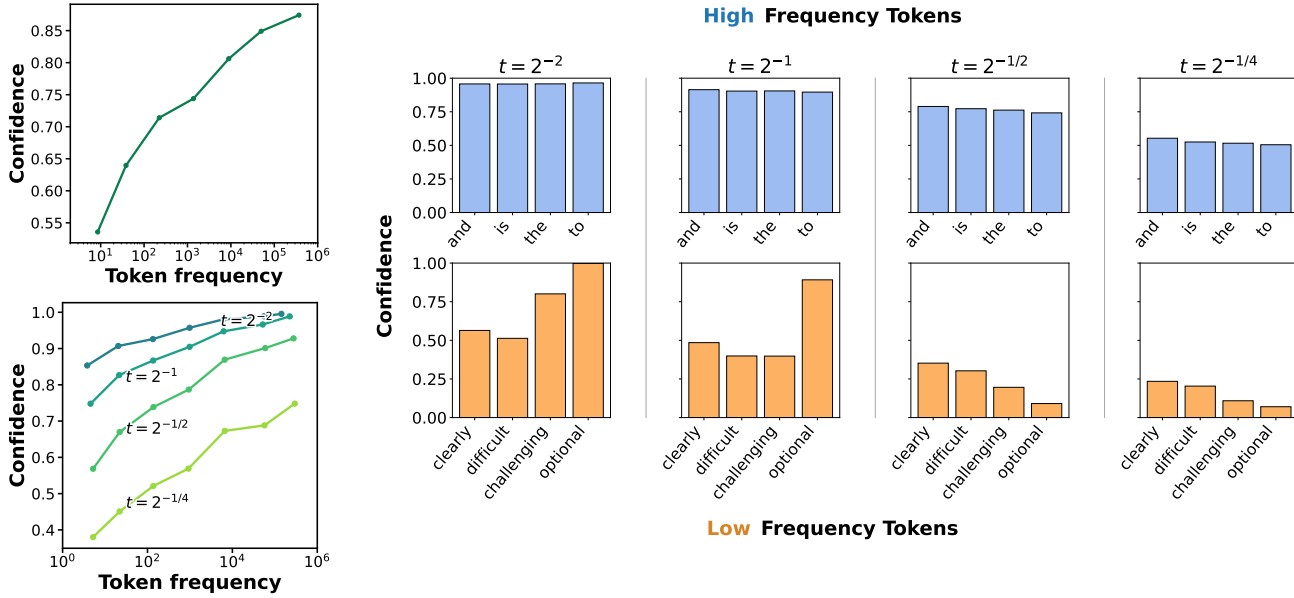

*(a)* Frequency vs. confidence.                    *(b)* Token-level confidence across timesteps.

*Figure 2.* **Token Analysis with LLaDA.** Using data collated from 4 post-training corpora (Muennighoff et al., 2025; Bercovich et al., 2025; Open-R1, 2025; Team OLMo et al., 2025), we analyze 0.5B masked tokens and aggregate token-level confidence and frequencies. **(a)** We bin tokens by log-scaled frequency and plot the mean model confidence against the average frequency. The marginalized plot (**top**) reveals that rare tokens have lower confidence on average, demonstrating that certain tokens are more difficult to predict (*what* dimension). We perform a more nuanced analysis by breaking down the marginalized plot by diffusion timestep $t$ (**bottom**), revealing an interaction between the *what* and *when* dimensions. Specifically, we observe a $t$-induced bias, when at large $t$ many of the model inputs are masked, low frequency tokens become disproportionately difficult to predict, suggesting that the information content of heavily masked inputs arising later in the forward diffusion process as diffusion time $t \to 1^{+}$ is insufficient to learn certain tokens reliably. Conversely, as $t \to 0^{-}$, less frequent tokens become more learnable, whereas predicting frequent tokens become trivial. **(b)** We sample representative high and low-frequency tokens, visualizing their (average) confidence across diffusion time. Rare tokens increasingly suffer as $t \to 1^{+}$, and experience more extreme drops in confidence than high frequency tokens.

*how can we design an SFT algorithm that accounts for them to effectively post-train DLMs?*

As our first contribution, we address (i) by analyzing SFT in DLMs and characterizing its failure cases. Specifically, we conduct an extensive analysis in Fig. 2a spanning 0.5B tokens collated from four popular post-training reasoning datasets (Muennighoff et al., 2025; Bercovich et al., 2025; Team OLMo et al., 2025; Open-R1, 2025). Across several pre-trained DLMs (Ye et al., 2025; Nie et al., 2025), our findings reveal two crucial considerations whose interplay govern SFT dynamics; those are, *what* tokens are learned, and *when* tokens are learned in the diffusion process. Our findings show that rare tokens in the corpus are more difficult to predict than frequent tokens (*what*). Additionally, rare tokens become more learnable when more context is available, corresponding to early forward diffusion times. However, at later forward diffusion times, the reduced information in the input disproportionately lowers the model's confidence on rare tokens, in some cases making them effectively unlearnable (*when*). These findings suggest that as forward diffusion time $t \to 1^{+}$, rare tokens often become unlearnable, making it more effective to focus compute

on frequent tokens. In contrast, as forward diffusion time $t \to 0^{-}$, frequent tokens are easy to predict, while rare tokens become more learnable. While prior works have proposed heuristics partially adhering to these guidelines by considering either the *what* or *when* dimensions in isolation (Ye et al., 2025; Xu et al., 2026), our study is the first to systematically analyze their combined effect during supervised fine-tuning. We show that modeling the interaction between token difficulty and diffusion time is critical for improving training.

As our second contribution, motivated by these insights, we propose and develop **LIFT**, the first post-training approach to target the interaction between *what* and *when* during DLM training. LIFT trains the model on masked tokens that are most appropriate to learn at each diffusion time given the available context. We obtain state-of-the-art results among various SFT training frameworks across two DLM base models on four reasoning benchmarks. We also evaluate LIFT on the challenging AIME-24 (AIME, 2024) and AIME-25 (Math-AI Team & Zhang, 2025), where it achieves up to a $3\times$ improvement over SFT baselines. Remarkably, LIFT attains performance close to the RLVR

baseline d1 (Zhao et al., 2025) while using roughly $500\times$ fewer GPU hours, establishing a new Pareto frontier for DLM post-training.

## 2. Related Work

**Diffusion Language Models** extend the success of diffusion models in continuous domains like image generation (Ho et al., 2020; Nichol & Dhariwal, 2021; Song & Ermon, 2019) to language. However, applying continuous diffusion to discrete text is inherently difficult (Austin et al., 2021a). To tackle this, Masked Diffusion Language Models (Sahoo et al., 2024) offer a discrete alternative by leveraging masked language modeling (Devlin et al., 2019), wherein tokens are randomly masked and the model learns to unmask them. Recent models (Nie et al., 2025; Ye et al., 2025) have shown competitive performance to autoregressive LLMs (ARMs) in mathematical reasoning, code generation (Zhu et al., 2025) and multi-modal tasks (Li et al., 2025), indicating that DLMs can perform complex reasoning. This makes DLM post-training a natural next step, with the goal of similar reasoning gains as in ARMs.

**Post-Training** of DLMs mirrors that of autoregressive models, following one of two approaches, namely reinforcement learning with verifiable rewards (RLVR) (Guo et al., 2025; Parashar et al., 2025; Zhao et al., 2025), or supervised fine-tuning (SFT). SFT with high-quality chain-of-thought data can achieve performance comparable to RL-based methods (Zelikman et al., 2022; Muennighoff et al., 2025). For DLMs, recent work with SFT has explored difficulty-informed training by considering *what* is being predicted (Li et al., 2025; Bie et al., 2025; Xu et al., 2026), since some tokens are inherently harder to predict, and *when* it is predicted (Ye et al., 2025), as inputs with heavier masking makes prediction more challenging. In this work, we investigate how jointly accounting for the interaction between *what* and *when* can improve the effectiveness of DLM post-training in enhancing reasoning performance.

## 3. Preliminaries

MDLMs (Sahoo et al., 2024; Nie et al., 2025; Ye et al., 2025) define a forward diffusion process on an input sequence $x_0$ from $p_{\text{data}}$, producing continuously indexed corrupted sequences $\{x_t\}_{t\in[0,1]}$ by progressively replacing tokens with *[MASK]*. The amount of information present in $x_t$ decreases monotonically with $t$ such that $x_1$ has all tokens masked. To generate a new sequence, MDLMs parameterize a bi-directional predictor $p_\theta$ to reverse the diffusion process starting from $x_1$. $p_\theta$ is trained by sampling a diffusion time $t \sim \pi(\cdot)$ with $t \in [0,1]$ (commonly $t \sim \text{Uniform}(0,1)$). To sample $x_t$, each token in $x_0$ is masked with probability $1 - \alpha_t$. Here, we follow the same setup as LLaDA (Nie

et al., 2025), wherein $\alpha_t = 1 - t$. Given the corrupted input $x_t$, $p_\theta$ learns to recover the original tokens from $x_0$ at the masked positions. The MDLM training objective is the negative evidence lower bound (NELBO) objective, which upper bounds the negative log-likelihood of the data. For a masked sequence $x_t$, the NELBO is given as

$$-\mathbb{E}_{t\sim\mathcal{U}[0,1],\, x_0\sim p_{\text{data}}} \left[ \frac{1}{t} \sum_{k=1}^{|x_0|} \mathbf{1}\{x_t^k = [MASK]\} \log p_\theta\left(x_0^k \mid x_t\right) \right] \tag{1}$$

where $|x_0|$ denotes the sequence length of $x_0$, $x_t^k$ is the token at position $k$ in the corrupted input, and $\mathbf{1}\{x_t^k = [MASK]\}$ restricts the loss to masked positions (predicting the corresponding $x_0^k$ given $x_t$). In vanilla SFT, the same loss is optimized directly on a supervised training set, with prompt tokens left unmasked.

## 4. Analysis

In this section we analyze token difficulty around the central question (Fig. 2a), *what* tokens should be learned and *when* in the diffusion process?

**Which tokens are difficult?** We investigate this question by analyzing denoising confidence, defined as the probability $p_\theta(x_0^k \mid x_t)$ assigned to the ground truth token $x_0^k$ at a masked position $k$, for a given noisy sequence $x_t$. Prior work in ARMs has shown that rare tokens, due to limited exposure during training, are harder to learn and consequently predict (Kandpal et al., 2023; Parashar et al., 2024; Udandarao et al., 2024). We test this in DLMs by masking inputs at random time steps (excluding prompt tokens) and measuring prediction confidence for the masked tokens. We then group tokens by their corpus frequency to analyze how difficulty varies with rarity.

**When do tokens become difficult to predict?** In DLMs, prediction difficulty depends not only on token identity but also on when the token is recovered during the denoising process. As the forward diffusion progresses, more of the input is masked, reducing the available context and making prediction harder. To analyze how difficulty evolves over time, we quantize the diffusion time $t$ into logarithmic bins ranging from $2^{-2}$ to $2^{-1/4}$ and measure average prediction confidence within each bin.

**Models and Datasets.** We conduct our analysis using two diffusion language models, LLaDA (Nie et al., 2025) and Dream (Ye et al., 2025), chosen for their differences in architecture and pre-training data. For the analysis, we use arithmetic reasoning post-training datasets that contain both questions and detailed reasoning traces: s1K (Muennighoff et al., 2025), the Nemotron Post-Training Dataset (Bercovich et al., 2025), Mixture of Thoughts (Open-R1, 2025), and

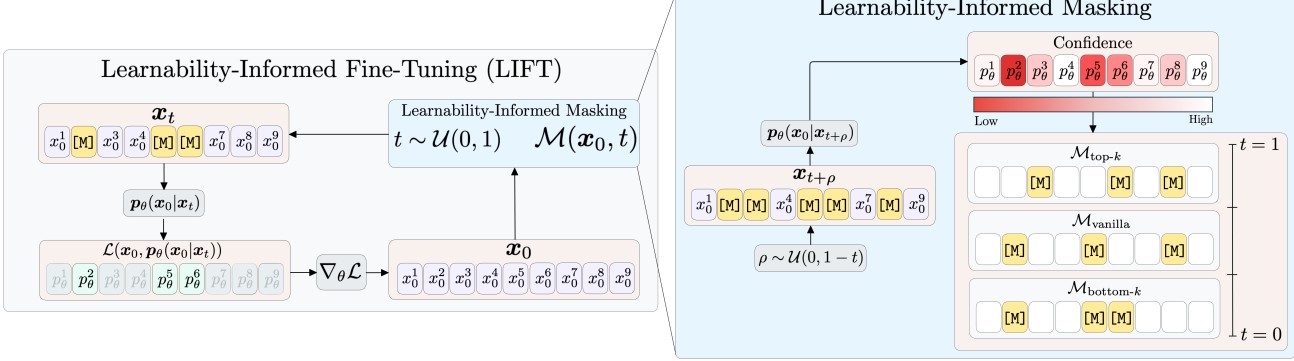

*Figure 3.* **Learnability-Informed Fine-Tuning (LIFT).** LIFT increases learnability by using model confidence and diffusion time to construct a learnability-informed mask so as to train on the highest utility tokens at each point in the diffusion process. Utility is estimated as a function of model confidence and diffusion time. In the first stage, a mask is sampled with rate $t + \rho$ and used to estimate model confidences $p_\theta(x_0 \mid x_{t+\rho})$ over all masked positions. LIFT then selects a subset of masked tokens from $x_{t+\rho}$ to supervise based on model confidences and diffusion time. Depending on the diffusion time, subset selection is either top-$K$ most confident tokens, bottom-$K$ least confident tokens, or vanilla (random). The mapping from diffusion time to subset selection method is done so as to increase learnability and utility of each training step according to the insights from our analysis in Sec. 4.

DociThink-RL (Team OLMo et al., 2025). Following the filtering procedure from Dream (Ye et al., 2025), we select examples where the combined length of the question and answer is less than 4096 tokens. This results in a dataset of approximately one million examples, totaling around 500 million tokens analyzed.

**Analysis Insights.** We first confirm that on average, rarer tokens are harder to predict than more frequent tokens (Fig. 2 a). Conditioning on diffusion time reveals a more nuanced pattern. As $t \to 0^-$, when substantial context remains unmasked, even rare tokens are comparatively easy to recover. As $t$ increases, the available information reduces. Beyond approximately $t \geq 2^{-1}$, prediction difficulty rises for all tokens, with rare tokens becoming the most challenging (Fig. 2). Overall, these results show that difficulty is jointly determined by *what* is being predicted (token frequency) and *when* it is predicted (diffusion time). This suggests that with the limited context accompanying $t \to 1^+$, model capacity and training iterations may not be optimally utilized by attempting to denoise rare tokens, and that efforts should instead be directed towards predicting tokens that are more feasible to learn. As information increases with decreasing $t$, rare tokens become more learnable, whereas the prediction of more frequent tokens is trivial and of limited benefit to train. We therefore propose to incorporate both dimensions so that training emphasizes targets that maximize learnability under the available context.

## 5. Methods

In this section, we present LIFT, a supervised fine-tuning method for efficient post-training of diffusion language models. LIFT is motivated by our analysis, where the difficulty

of predicting tokens depends on the interaction between *what* and *when*, i.e., token frequency and the amount of unmasked tokens available in the input. Following this principle, LIFT adaptively selects which tokens to learn at each timestep, focusing on easy and frequent tokens when the input is heavily masked, and on rare and difficult tokens when more context is available. This enhances the information gained in each training step by simultaneously ensuring that target tokens are learnable and are non-trivial to predict.

**Which tokens to select for training?** Instead of training directly on the input $x_t$ randomly masked at timestep $t$, LIFT applies **learnability-informed masking** to maximize the learning signal of training targets. This is done by first sampling a secondary masking ratio $\rho \sim \mathcal{U}(0, 1 - t)$ to construct a more corrupted input $x_{t+\rho}$ from which learnability can be estimated. For example, if $t = 0.4$ and $\rho = 0.3$, we create $x_{t+\rho}$ where 70% of the tokens are masked. Having created $x_{t+\rho}$, we obtain confidence scores for the ground truth token $c = p_\theta(x_0^k | x_{t+\rho})$ at each masked position. We define token difficulty simply as the corresponding loss, $\ell_k = -\log c_k$, where lower confidence naturally indicates a harder token.

LIFT then constructs a learnability-informed mask by selecting a subset of the masked tokens in $x_{t+\rho}$ to supervise dependent on diffusion time, e.g., the top-$K$ tokens (highest confidence, easy tokens) or the bottom-$K$ tokens (lowest confidence, hard tokens), where $K = t \cdot |x_0|$. The remaining masked positions, which are not selected for training, are filled in using the original tokens from the clean input $x_0$, giving us $x_t$. LIFT then uses $x_t$ as input to $p_\theta$ for computing the NELBO in Eq. (1).

---

**Algorithm 1** LIFT: Learnability-Informed Fine-Tuning

---

**Require:** Dataset $p_{\text{data}}$, parameter $H \geq 2$, chosen variant, (LIFT or LIFT-A), learning rate $\eta$

1: **repeat**
2:    $x_0 \sim p_{\text{data}}, \quad t \sim \mathcal{U}[0,1], \quad \rho \sim \mathcal{U}[0, 1-t]$           ▷ Sample input, timestep, and secondary ratio.
3:    $x_{t+\rho} \sim q(x_{t+\rho} \mid x_0), \quad c_k \leftarrow p_\theta(x_0^k \mid x_{t+\rho}) \quad \forall k \in \mathcal{M}_{t+\rho}$    ▷ Mask input and compute confidences.
4:    $\mathcal{S}_t \leftarrow$ Eq. (2) with $K = \lfloor t \cdot |x_0| \rfloor$           ▷ Select tokens to supervise.
5:    **if** LIFT **then**
6:       $x_t \leftarrow x_{t+\rho}$           ▷ Create $x_t$ based on learnability.
7:       **for** $k \in \mathcal{M}_{t+\rho} \setminus \mathcal{S}_t$ **do**
8:          $x_t^k \leftarrow x_0^k$           ▷ Unmask unsupervised masked tokens.
9:       **end for**
10:   **else if** LIFT-A **then**
11:      $x_t \leftarrow x_{t+\rho}$
12:      $t \leftarrow t + \rho$
13:   **end if**
14:   $\theta \leftarrow \theta - \eta \nabla_\theta \left[ -\frac{1}{t} \sum_{k \in \mathcal{S}_t} \log p_\theta(x_0^k \mid x_t) \right]$           ▷ Take gradient descent step.
15: **until** converged

---

**When should tokens be learned?** As demonstrated in Sec. 4, learnability of tokens is dependent on token difficulty and the amount of context available, i.e., the proportion of unmasked tokens, which is a function of diffusion time. When $t \to 1^+$, the hardest tokens to predict can be unlearnable due to insufficient information, whereas $(t \to 0^-)$, the easiest tokens are trivial to predict. Both cases are undesirable, as neither provide high-utility learning signal.

LIFT addresses this by selecting the subset of masked tokens from $x_{t+\rho}$ according to the diffusion time. Let $\mathcal{M}_t$ and $\mathcal{M}_{t+\rho}$ denote the sets of masked token indices at times $t$ and $t+\rho$, respectively. We define operators $\text{Top}_K(\mathcal{S}, c)$ and $\text{Bottom}_K(\mathcal{S}, c)$ that return the subset of $K$ indices from a set $\mathcal{S}$ corresponding to the highest and lowest confidence scores $c$, respectively. To control the scheduling behavior, we introduce a new parameter $H \geq 2$ that partitions $t$ into three regimes to define the selected subset for supervision, denoted $\mathcal{S}_t \subseteq \mathcal{M}_{t+\rho}$:

$$\mathcal{S}_t = \begin{cases} \text{Bottom-}K(\mathcal{M}_{t+\rho}, c) & \text{if } t \in \left(0, \frac{1}{H}\right) \\ \mathcal{M}_t & \text{if } t \in \left[\frac{1}{H}, 1-\frac{1}{H}\right) \\ \text{Top-}K(\mathcal{M}_{t+\rho}, c) & \text{if } t \in \left[1-\frac{1}{H}, 1\right] \end{cases} \quad (2)$$

This selection reflects the insights drawn from our analysis that when the input has many masked positions $(t \to 1^+)$, we train on easy tokens using Top-$K$, where learnability is highest despite limited context. When corruption is moderate, we revert to standard vanilla SFT. When the input has low corruption $(t \to 0^-)$, we learn the hardest tokens using Bottom-$K$. This ensures that tokens are learned when they are most appropriate to learn, based on the level of context available at each timestep. By replacing the standard

masking indicator with $\mathbf{1}\{k \in \mathcal{S}_t\}$, the modified NELBO restricts the loss exclusively to this subset:

$$\mathcal{L}_{\text{LIFT}} = -\mathbb{E}_{\substack{t \sim \mathcal{U}[0,1] \\ x_0 \sim p_{\text{data}}}} \left[ \frac{1}{t} \sum_{k=1}^{|x_0|} \mathbf{1}\{k \in \mathcal{S}_t\} \log p_\theta(x_0^k \mid x_t) \right]$$
$$(3)$$

In our experiments, we find that integer values $H = 2$ or $H = 3$ work well in practice. As $H$ increases beyond 3, LIFT behaves increasingly like vanilla SFT, since the middle region $\left[\frac{1}{H}, 1-\frac{1}{H}\right]$ dominates the training.

**Approximate Variant of LIFT.** Since LIFT selects tokens for training based on confidence under the model $p_\theta$, it requires two forward passes, one to obtain token confidences $p_\theta(x_0^k | x_{t+\rho})$, and another $p_\theta(x_0^k | x_t)$ to compute the final loss. To reduce this computational overhead, our lightweight variant, LIFT-A, performs only a single forward pass at $t + \rho$ and applies a gating mask that zeroes out the loss for tokens not selected for supervision (i.e., those outside $\mathcal{S}_t$). Because the loss is evaluated at $t+\rho$ rather than the true diffusion timestep $t$, this objective represents a biased NELBO. This approximation trades off loss accuracy for efficiency by calculating the loss at $t + \rho$ and avoiding a second forward pass at $t$.

$$\mathcal{L}_{\text{LIFT-A}} = -\mathbb{E}_{\substack{t \sim \mathcal{U}[0,1] \\ \rho \sim \mathcal{U}[0,1-t] \\ x_0 \sim p_{\text{data}}}} \left[ \frac{1}{t + \rho} \sum_{k=1}^{|x_0|} \mathbf{1}\{k \in \mathcal{S}_t\} \log p_\theta(x_0^k \mid x_{t+\rho}) \right]$$
$$(4)$$

**Connection to Curriculum Learning.** While our method is motivated by the notion of token difficulty, it does not follow the conventional curriculum learning (Bengio et al., 2009) where data is presented in an increasing order of difficulty. Instead, **LIFT adaptively performs token selection**

**based on learnability**, accounting for both the available unmasked context at each timestep and the model's improving capacity throughout training. However, recent work has shown that curriculum learning and adaptive sampling can offer complementary benefits (Parashar et al., 2025; Yu et al., 2025; Chen et al., 2025), and future work could explore integrating the two.

# 6. Experiments

In this section, we evaluate LIFT on a suite of mathematical reasoning tasks spanning a range of difficulty levels. We demonstrate that LIFT consistently outperforms all baseline methods. The results indicate that difficulty-informed training of LIFT is a simple yet effective approach for SFT-based post-training of diffusion language models. We begin by describing the datasets, baseline methods. We then explain the evaluation metrics and main experimental results followed by detailed ablations. We include the training implementation details in the Appendix.

## 6.1. Setup

**Training Datasets.** We use s1K (Muennighoff et al., 2025), which comprises 1,000 high-quality chain-of-thought (CoT) traces generated by Gemini. Prior work has shown the effectiveness of supervised fine-tuning on s1K (Xu et al., 2026; Zhao et al., 2025), making it a strong baseline for comparison. To explore the effect of varying fine-tuning data on training, we also construct a larger dataset of approximately 12,000 problems by randomly sampling the collated datasets used in our analysis, namely, Nemotron Post-training Dataset (Bercovich et al., 2025), Mixture of Thoughts (Open-R1, 2025), and DociThink R1 (Team OLMo et al., 2025). We refer to this dataset as LIFT-SFT-12K. While s1K (Muennighoff et al., 2025) is a highly curated dataset with clean, expert-crafted CoT traces, LIFT-SFT-12K is less specialized and has a more heterogeneous training distribution.

**Evaluation.** We follow the evaluation setup of d1 (Zhao et al., 2025) and assess LIFT on GSM8K (Cobbe et al., 2021), MATH (Hendrycks et al., 2021), Countdown (Gandhi et al., 2024), and Sudoku (Cordero). We use the same evaluation code, prompts, and inference settings as d1, and report accuracy (pass@1). In addition, we evaluate on AIME'24 (AIME, 2024) and AIME'25 (Math-AI Team & Zhang, 2025) datasets to measure LIFT on advanced mathematical reasoning; given their difficulty, we report pass@16 for AIME. We include pass@8 and avg@8, and avg@16 results in the Appendix (see Table 12).

**Baselines.** Since we fine-tune LLaDA Instruct (Nie et al., 2025) and LLaDA 1.5 (Zhu et al., 2025), they are our first set of baselines. We use the vanilla masked-DLM objective (Sahoo et al., 2024) (**Vanilla**). We additionally consider the methods of Xu et al. (2026) and (Ye et al., 2025). Context-Adaptive noise Rescheduling at Token-level (**CART**) (Ye et al., 2025) re-weights each masked token in the NELBO objective such that targets with fewer unmasked tokens in their immediate neighborhood have less weight, as these tokens are harder to denoise. This accounts for variable amount of context across diffusion time (*when*), however, it is applied independetly of token identity, and thus, does not consider *what*. Guided Importance-Aware Fine-Tuning (**GIFT**) (Xu et al., 2026) instead accounts for *what* without *when*. Similar to LIFT, GIFT estimates token-level uncertainty using an initial forward pass of the model with all non-prompt tokens masked as $p_\theta(\cdot|x_1)$. Each response token is then masked with probability proportional to the square root of the token-level entropy such that tokens with high uncertainty are more likely to be masked. While this shares connections with the bottom-$K$ loss, it is independent of time, since uncertainty is always estimated conditioned on $x_1$. The inclusion of both GIFT and CART serves to compare the effectiveness of jointly accounting for the interaction between *what* and *when* dimensions as done by LIFT compared to modeling only one dimension in isolation.

## 6.2. Results

We now present the results of LIFT, reporting the mean performance across three training runs with different random seeds. The same procedure is applied to all baselines in Table 1. Additionally, we highlight the relative gains over the base model in green. Confidence intervals are included in the Appendix B.2.

**LIFT consistently outperforms baselines across both LLaDA-8B-Instruct and LLaDA 1.5.** Table 1 presents the performance of LIFT with two values of $H$, namely 2 and 3, denoted as $\text{LIFT}_2$ and $\text{LIFT}_3$, respectively. On LLaDA-8B-Instruct, our method shows notable improvements over the baseline, especially on harder benchmarks AIME 2024 and 2025, where LIFT improves base model performance by more than $2\times$. Finally, we find that $\text{LIFT}_3$ offers more consistent improvements across benchmarks and base models compared to $\text{LIFT}_2$.

**Training Distribution Robustness of LIFT.** To assess the generality of LIFT to different fine-tuning datasets, we conduct experiments on the LIFT-SFT-12K dataset described in Sec. 6.1. As shown in Table 2, LIFT demonstrates consistent gains across evaluation tasks, indicating that its effectiveness is not limited to s1K (Muennighoff et al., 2025). These results suggest that LIFT generalizes well and could serve as a scalable objective beyond supervised post-training and could potentially be useful for broader pre-training or

*Table 1.* **LIFT outperforms baselines on LLaDA-8B-Instruct and LLaDA-1.5.** Across 4 math and reasoning benchmarks (Cobbe et al., 2021; Hendrycks et al., 2021; Gandhi et al., 2024; Cordero), LIFT with $H \in \{2, 3\}$ outperforms post-training baselines Vanilla SFT, GIFT (Xu et al., 2026), and CART (Ye et al., 2025). Additionally, LIFT demonstrates $3\times$ relative gain in pass@16 accuracy with LLaDA on AIME'24 (AIME, 2024) and AIME'25 (Math-AI Team & Zhang, 2025). Percent deltas denote relative change versus the corresponding pre-trained model.

| Name | GSM8K | MATH | Countdown | Sudoku | AIME '24 | AIME '25 |
|---|---|---|---|---|---|---|
| **LLaDA** | | | | | | |
| LLaDA | 78.1 | 36.1 | 19.6 | 11.2 | 3.3 | 3.3 |
| Vanilla | 78.7 | 34.1 | 20.7 | 16.8 | 6.7 | 3.3 |
| GIFT | 79.2 | 34.2 | 21.7 | 17.3 | 16.7 | 0.0 |
| CART | 78.8 | 35.5 | 23.0 | 14.6 | 10.0 | 3.3 |
| **LIFT$_2$** | 79.8 | 37.9 | 27.9 | 16.5 | 10.0 | 3.3 |
| | ↑ **2.1%** | ↑ **4.9%** | ↑ **42.3%** | ↑ **47.3%** | ↑ **203.0%** | ↑ 0.0% |
| **LIFT$_3$** | 79.4 | 38.4 | 26.4 | 17.4 | 16.7 | 6.7 |
| | ↑ **1.6%** | ↑ **6.3%** | ↑ **34.6%** | ↑ **55.3%** | ↑ **406.0%** | ↑ **103.0%** |
| **LLaDA-1.5** | | | | | | |
| LLaDA 1.5 | 80.9 | 37.8 | 22.6 | 12.1 | 13.3 | 3.3 |
| Vanilla | 79.2 | 32.6 | 22.0 | 14.4 | 6.7 | 3.3 |
| GIFT | 79.5 | 36.0 | 20.7 | 17.6 | 6.7 | 3.3 |
| CART | 80.4 | 35.8 | 21.5 | 17.0 | 6.7 | 0.0 |
| **LIFT$_2$** | 79.5 | 39.8 | 31.3 | 15.6 | 13.3 | 6.7 |
| | ↓ 1.7% | ↑ **5.2%** | ↑ **38.4%** | ↑ **28.9%** | ↑ 0.0% | ↑ **103.0%** |
| **LIFT$_3$** | 82.2 | 38.8 | 31.2 | 18.2 | 13.3 | 6.7 |
| | ↑ **1.6%** | ↑ **2.6%** | ↑ **38.0%** | ↑ **50.4%** | ↑ 0.0% | ↑ **103.0%** |

*Table 2.* **LIFT is robust to training datasets.** Benchmark performance when training on LIFT-SFT-12K, a math-focused dataset assembled by randomly sampling from multiple post-training sources. LIFT consistently improves performance, demonstrating strong generalization across training datasets.

| Name | GSM 8K | MATH | Count down | Sudoku | AIME '24 | AIME '25 |
|---|---|---|---|---|---|---|
| Instruct | 78.2 | 36.8 | 20.0 | 11.8 | 3.3 | 3.3 |
| Vanilla | 82.9 | 34.6 | 19.9 | 9.3 | 3.3 | 3.3 |
| GIFT | 82.4 | 34.4 | 25.0 | 6.6 | 6.7 | 0.0 |
| CART | 80.0 | 34.6 | 24.6 | 11.6 | 6.7 | 0.0 |
| **LIFT$_2$** | 81.8 | 38.0 | 25.8 | 12.5 | 6.7 | 3.3 |
| | ↑ **4.6%** | ↑ **3.2%** | ↑ **29%** | ↑ **5.9%** | ↑ **103.0%** | ↑ 0.0% |
| **LIFT$_3$** | 81.4 | 38.6 | 20.7 | 10.3 | 10.0 | 3.3 |
| | ↑ **4.1%** | ↑ **4.9%** | ↑ **3.5%** | ↓ 12.7% | ↑ **203.0%** | ↑ 0.0% |

*Table 3.* **Compute–performance trade-off.** We compare methods using H100 GPU hours alongside benchmark performance, including an RLVR oracle (d1), and the single-forward-pass approximation LIFT-A. LIFT (and LIFT-A) delivers substantial gains at much lower compute.

| Name | H100 Hours | GSM 8K | MATH | Count down | Sudoku | AIME '24 | AIME '25 |
|---|---|---|---|---|---|---|---|
| Vanilla | 1.0 | 78.7 | 34.1 | 20.7 | 16.8 | 6.7 | 3.3 |
| CART | 1.0 | 78.8 | 35.5 | 23.0 | 14.6 | 10.0 | 3.3 |
| **LIFT$_2$-A** | 1.0 | 78.7 | 36.8 | 33.2 | 11.1 | 6.7 | 3.3 |
| **LIFT$_3$-A** | 1.0 | 79.0 | 34.0 | 23.1 | 16.2 | 13.4 | 3.3 |
| GIFT | 1.8 | 79.2 | 34.2 | 21.7 | 17.3 | 16.7 | 0.0 |
| **LIFT$_2$** | 1.8 | 79.8 | 37.9 | 27.9 | 16.5 | 10.0 | 3.3 |
| **LIFT$_3$** | 1.8 | 79.4 | 38.4 | 26.4 | 17.4 | 16.7 | 6.7 |
| d1 (oracle) | 2303 | 81.9 | 39.2 | 37.1 | 18.4 | — | — |

instruction tuning settings.

**Compute–Performance Trade-offs and the Pareto Frontier.** Table 3 presents results for LIFT-A on LLaDA 8B-Instruct, a compute-efficient variant that requires only a single forward pass. Despite its lower computational cost, LIFT-A consistently outperforms baselines with comparable budgets, such as vanilla and CART, highlighting a favorable trade-off between efficiency and performance.

Remarkably, when compared to d1 (Zhao et al., 2025), a reinforcement learning-based post-training method that fine-tunes separately for each task and requires over 2,000 H100 GPU hours, LIFT achieves similar performance while using only 1.8 H100 hours (Table 3). This $1000\times$ reduction in compute demonstrates the strength of our learnability-informed training approach in losslessly enhancing training efficiency. As illustrated in Figure 4, LIFT establishes a new compute-efficient Pareto frontier. These findings suggest that while RL has been effective for ARMs (Guo et al.,

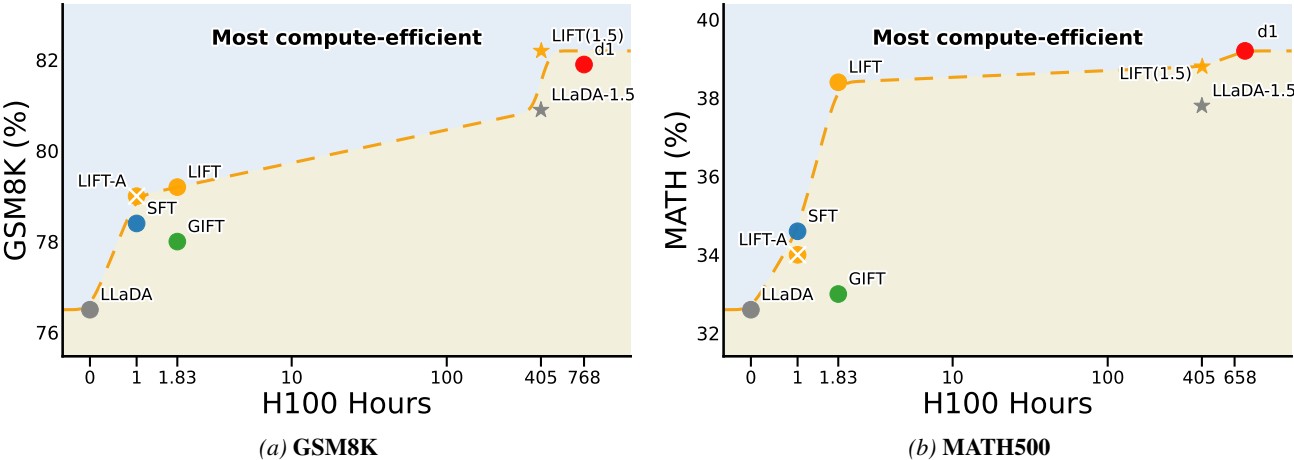

*(a)* **GSM8K**                                                    *(b)* **MATH500**

*Figure 4.* **LIFT lies on the compute-efficient Pareto frontier**, measured in H100 GPU hours. When applied to LLaDA, LIFT requires only 2 hours of training and already outperforms baselines on GSM8K and MATH. We also evaluate LIFT-A, an approximate variant of our method, which performs comparably at half the compute budget of LIFT. Finally, when LIFT is applied to LLaDA 1.5, which requires approximately 405 H100 hours of pretraining, LIFT(1.5) adds just 2 hours, performing similar on MATH and outperforming d1 (Zhao et al., 2025) on GSM8K, while using nearly 50% less total compute.

*Table 4.* **LIFT is robust across generation lengths.** We follow evaluation setup of d1 (Zhao et al., 2025) and compare performance across generation lengths of 128, 256, and 512 tokens on different datasets. LIFT is robust across lengths and generally benefits from longer generations, except on Sudoku. Best results are in **bold**.

| Name | GSM8K | | | MATH | | | Countdown | | | Sudoku | | |
|---|---|---|---|---|---|---|---|---|---|---|---|---|
| | **128** | **256** | **512** | **128** | **256** | **512** | **128** | **256** | **512** | **128** | **256** | **512** |
| Instruct | 68.5 | 76.1 | 78.1 | 26.4 | 32.4 | 36.1 | 19.6 | 19.6 | 17.1 | 11.2 | 6.5 | 5.5 |
| Vanilla | 67.1 | **78.5** | 78.7 | 27.0 | 32.8 | 34.1 | 20.1 | 16.2 | 20.7 | 16.8 | 7.3 | 4.7 |
| GIFT | 66.4 | 78.0 | 79.2 | 27.2 | 32.7 | 34.2 | 21.7 | 16.4 | 17.3 | 16.0 | **8.7** | 5.2 |
| CART | 67.2 | 76.6 | 78.9 | 24.9 | 30.5 | 35.5 | **23.0** | 19.0 | 18.7 | 14.6 | 8.5 | 4.7 |
| **LIFT$_2$** | **70.9** | 78.2 | **79.8** | **29.0** | 35.5 | 37.9 | 22.8 | 17.9 | **28.0** | 16.5 | 7.2 | 6.9 |
| **LIFT$_3$** | 69.5 | 78.4 | 79.4 | 28.2 | **37.6** | **38.4** | 20.1 | **21.3** | 26.4 | **17.4** | 7.9 | **8.5** |

2025), efficient RL-based post-training for DLMs remains an open challenge.

## 6.3. Ablation Studies

To better understand the design and performance of LIFT, we conduct a series of ablations focusing on its key components. All ablations were carried out by training LLaDA-8B-Instruct on s1K.

**Ablation on Generation Length.** Following the setup used in d1 and LLaDA (Zhao et al., 2025; Nie et al., 2025), we ablate the response length across 128, 256, and 512 tokens, with diffusion steps set to half the generation length. Results are shown in Table 4, and we report the best-performing value in all main tables, consistent with prior work (Zhu et al., 2025; Xu et al., 2026). LIFT performs robustly across lengths, with performance generally improving as generation length increases, except on Sudoku, which exhibits the opposite trend.

**Ablating the Interaction between What and When.** LIFT builds directly on our analysis, where we demonstrated the substantial effect of the interaction between *what* and *when* on the loss landscape of DLMs. We next study this key design choice by considering ablated versions of LIFT that only account for *what* tokens are learned without any constraint on *when* they are learned in the diffusion process. To construct frameworks that only consider *what* is learned and are independent of diffusion time, we introduce bottom-$K$ and top-$K$ training as standalone baselines. Bottom-$K$ trains only on hard tokens, while top-$K$ focuses only on easy ones. Alternatively, to analyze whether the mixture of top and bottom $K$ losses is sufficient without consideration of *when* these losses are applied in the diffusion process, we design a time-independent variant of LIFT by randomly selecting one of bottom-$K$, vanilla, or top-$K$ losses at each training step. We refer to these as Random$_2$ and Random$_3$, where Random$_2$ samples between bottom-$K$ and top-$K$, and Random$_3$ samples from all three.

*Table 5.* **Ablation of interaction between *what* and *when*.** To ablate the importance of *what*, we introduce Top-$K$ and Bottom-$K$ as baselines, which train on the most and least confident masked tokens, respectively. Furthermore we ablate the time-independent variant of LIFT by randomly selecting one of Top-$K$, Bottom-$K$ ($\text{Random}_2$) and additionally Vanilla ($\text{Random}_3$). As seen below, improvements from these baselines is not consistent across tasks. By accounting for both *what* and *when*, LIFT achieves robust performance across all tasks, empirically validating the consideration of both *what* and *when* during SFT training of DLMs.

| Name | GSM 8K | MATH | Count down | Sudoku | AIME '24 | AIME '25 |
|---|---|---|---|---|---|---|
| Vanilla | 78.7 | 34.1 | 20.7 | 16.8 | 6.7 | 3.3 |
| Top-$K$ | 77.2 | 34.6 | 30.3 | 18.0 | 3.3 | 0.0 |
| Bottom-$K$ | 77.5 | 34.8 | 26.0 | 18.0 | 10.0 | 3.3 |
| $\text{Random}_2$ | 80.1 | 37.8 | 23.0 | 16.8 | 0.0 | 0.0 |
| $\text{Random}_3$ | 80.0 | 35.8 | 23.4 | 18.0 | 0.0 | 0.0 |
| $\textbf{LIFT}_2$ | 79.8 | 37.9 | 28.0 | 16.5 | 10.0 | 3.3 |
| $\textbf{LIFT}_3$ | 79.4 | 38.4 | 26.4 | 17.4 | 16.7 | 6.7 |

*Table 6.* **Ablations of H for LIFT.** We ablate the value of H, which controls whether Top-$K$, Bottom-$K$, or Vanilla SFT is applied during training. Mathematically, as $H \to \infty$, LIFT converges to vanilla SFT. Empirically, $H = 3$ achieves the best average performance across benchmarks.

| Name | GSM 8K | MATH | Count down | Sudoku | AIME '24 | AIME '25 |
|---|---|---|---|---|---|---|
| Vanilla | 78.7 | 34.1 | 20.7 | 16.8 | 6.7 | 3.3 |
| $\textbf{LIFT}_2$ | 79.8 | 37.9 | 28.0 | 16.5 | 10.0 | 3.3 |
| $\textbf{LIFT}_3$ | 79.4 | 38.4 | 26.4 | 17.4 | 16.7 | 6.7 |
| $\textbf{LIFT}_4$ | 78.0 | 37.2 | 30.4 | 14.9 | 6.7 | 3.3 |
| $\textbf{LIFT}_5$ | 78.2 | 35.4 | 22.7 | 15.7 | 6.7 | 3.3 |
| $\textbf{LIFT}_{10}$ | 78.2 | 34.2 | 21.7 | 16.1 | 6.7 | 3.3 |
| $\textbf{LIFT}_{15}$ | 77.8 | 34.1 | 22.6 | 15.8 | 3.3 | 3.3 |
| $\textbf{LIFT}_{20}$ | 78.0 | 33.4 | 21.0 | 15.5 | 6.7 | 3.3 |

Results for this ablation are shown in Tab. 5. With the except of Countdown and Sudoku, LIFT offers substantial gains over both the Top and Bottom-$K$ loss variants. While the Random variants are competitive with LIFT across benchmarks, they achieve Pass@16 of 0 on the challenging AIME benchmarks, suggesting that accounting for the interaction of *what* tokens are learned *when* is crucial for success in real-world tasks requiring multi-step reasoning and use of tokens that are in the tails of the base model distribution.

**Ablation of $H$.** We ablate the hyperparameter $H$ in LIFT, which determines the rate at which top-$K$, bottom-$K$, or vanilla is used during training (See Sec 5). Mathematically, as $H \to \infty$, LIFT approaches vanilla SFT. As shown in Table 6, LIFT is robust to this parameter, with $H = 3$ yielding the best average performance across benchmarks.

**Extension to Dream-7B** To further evaluate the generalizability of LIFT, we extended LIFT to Dream (Ye et al., 2025). As demonstrated in Table 7, LIFT yields perfor-

*Table 7.* **Extension of LIFT to Dream-7B** (Ye et al., **2025**). LIFT demonstrates robust performance gains across mathematical and reasoning benchmarks.

| Method | GSM8K | MATH | Countdown | Sudoku |
|---|---|---|---|---|
| Instruct | 76.7 | 39.8 | 21.1 | 8.2 |
| Vanilla | 76.1 | 30.6 | 25.0 | 14.8 |
| GIFT | 78.5 | 40.0 | 23.4 | 16.0 |
| CART | 77.8 | 38.9 | 22.3 | 17.2 |
| $\textbf{LIFT}_2$ | 77.9 | **40.8** | **33.6** | **22.5** |
| $\textbf{LIFT}_3$ | **79.1** | 40.6 | 25.6 | 17.5 |

*Table 8.* **Ablation of alternative sampling strategies for $\rho$.** We compare the uniform sampling of $\rho \sim \mathcal{U}(0, 1-t)$ in LIFT against fixed schedules ($\rho = \min(k, 1-t)$) and variance-reduced distributions ($\rho \sim \mathcal{U}(k, 1-t)$). We experimented with bounds $k \in \{0.1, 0.3\}$ for a maximum generation length of 256 tokens.

| Strategy | GSM8K | MATH | Countdown | Sudoku |
|---|---|---|---|---|
| $\rho = \min(0.1, 1-t)$ | 78.2 | 36.4 | 21.1 | 7.6 |
| $\rho = \min(0.3, 1-t)$ | 77.9 | 34.6 | 21.1 | **8.5** |
| $\rho \sim \mathcal{U}(1-t, 0.1)$ | 78.9 | 34.4 | 20.0 | 4.5 |
| $\rho \sim \mathcal{U}(1-t, 0.3)$ | 78.4 | 36.2 | 22.8 | 6.5 |
| $\textbf{LIFT}_3$: $\rho \sim \mathcal{U}(0, 1-t)$ | **79.4** | **37.6** | **26.4** | 7.9 |

mance gains consistent with other models.

**Alternate sampling strategies for $\rho$** To evaluate the impact of alternative sampling strategies for $\rho$, we a fixed schedule ($\rho = \min(k, 1-t)$) and variance-reduced uniform distribution ($\rho \sim \mathcal{U}(k, 1-t)$). As shown in Table 8, our default approach, i.e., ($\rho \sim \mathcal{U}(0, 1-t)$), performs best. By avoiding the deterministic constraints of the fixed schedule and the truncated intervals of the variance-reduced distributions, uniform sampling of $\rho$ maximizes the diversity of masking patterns the model encounters during training. During fine-tuning, this diversity acts as implicit data augmentation, mirroring an effect previously observed in image diffusion (Kingma et al., 2021).

## 7. Conclusion

We propose LIFT, a learnability-informed fine-tuning method for post-training DLMs. LIFT builds on the insight that certain tokens are inherently harder to learn (*what*), and that their learnability depends on *when* they are predicted during the diffusion process. To elucidate this relationship, we analyze over 0.5B tokens across common post-training datasets, revealing consistent patterns in token frequency and the dependence on diffusion timestep. These findings inform the design of LIFT, which achieves state-of-the-art performance across arithmetic reasoning tasks, with particularly strong gains on challenging benchmarks such as AIME. Notably, LIFT establishes a new compute-efficient Pareto frontier, matching the performance of RL-based methods while requiring orders of magnitude less compute.

## Impact Statement

This paper presents work whose goal is to advance the field of Machine Learning. There are many potential societal consequences of our work, none which we feel must be specifically highlighted here.

## Acknowledgments

This work was supported in part by ARPA-H under grant 1AY1AX000053, NIH under grant U01AG070112, and NSF under grant CNS-2328395.

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

# Learnability-Informed Fine-Tuning of Diffusion Language Models
## Appendix

## A. Pareto Frontier for Countdown and Sudoku

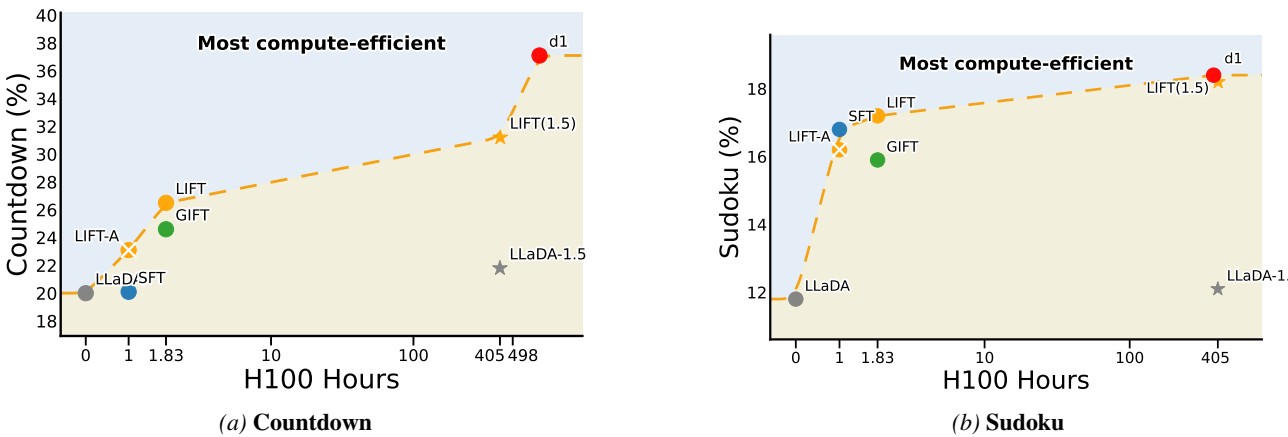

*(a)* **Countdown**

*(b)* **Sudoku**

*Figure 5.* **Accuracy vs. H100 hours (log scale)** across Countdown, and Sudoku.

We show the pareto frontier for Countdown and Sudoku in Fig 5.

## B. Additional Analysis and Ablations

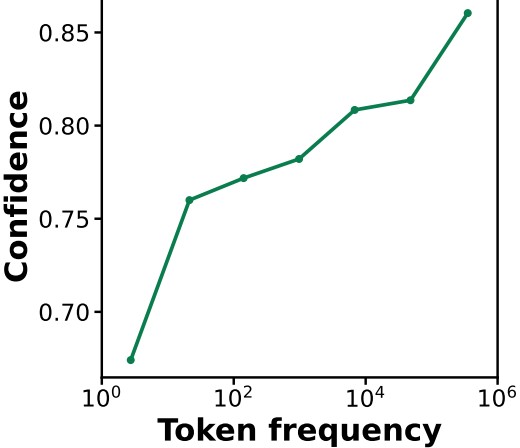

*(a)* **Dream confidence vs. token frequency (global).**

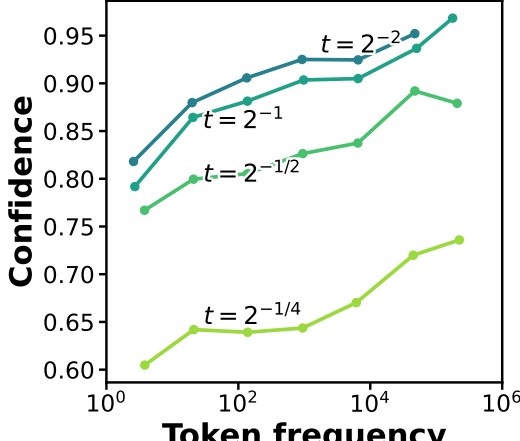

*(b)* **Dream confidence vs. token frequency (timestep separated).**

*Figure 6.* **Dream Token Analysis.** For each token, we compute Dream's mean confidence when the token is the masked target and plot it against the token's frequency in our collated post-training corpus. To reduce noise, tokens are grouped into shared log-spaced frequency bins (with a final tail bin for the most frequent tokens), and we plot the bin-wise average confidence versus the bin's mean frequency. We show the marginalized global trend (left) and the same relationship stratified by diffusion timestep (right). This was done on a scale of 1.22e8 tokens.

### B.1. Dream Token Analysis

The analysis visualized in Figure 2 is extended to other DLMs. We analyze masked token frequencies and confidences for Dream (Ye et al., 2025) in Figure 6. On average, the same trend is realized; at higher timesteps, the confidence distribution

favors frequent tokens.

Additionally, we sample tokens from each frequency bin in s1K and visualize them alongside the corresponding average confidence that LLaDA exhibits for tokens in that bin in Table 9. Again, we observe a clear frequency–confidence trend: high-frequency tokens are associated with higher average confidence, while rare tokens tend to receive lower confidence, consistent with the patterns in our aggregate plots.

*Table 9.* Word clouds of sampled tokens from s1K within each frequency bin, alongside the average LLaDA confidence computed over *all* tokens in that bin.

| Frequency Bin | Tokens | Confidence |
|---|---|---|
| $10^1$–$10^2$ |  | 0.6754 |
| $10^2$–$10^3$ |  | 0.7270 |
| $10^3$–$10^4$ |  | 0.7788 |
| $10^4$–$10^5$ |  | 0.8431 |
| $10^5+$ |  | 0.8829 |

## B.2. Confidence Intervals

We report the confidence intervals for our experiments on different datasets.

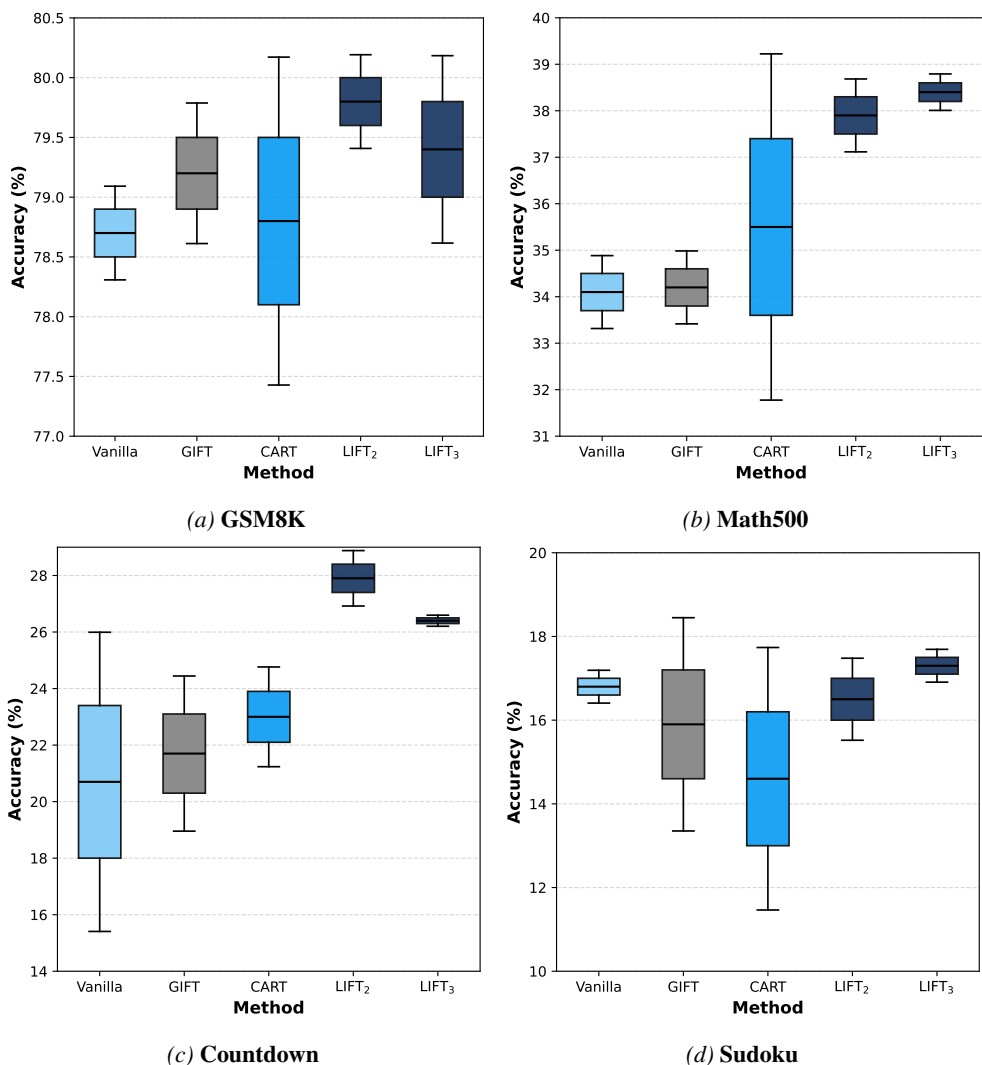

*Figure 7.* **Confidence Intervals for our experiments obtained via three runs on different separate seeds.** The box plots illustrate the distribution of accuracy scores over multiple seeds for five experimental methods. The central horizontal lines represent the median, while the box and whiskers quantify the confidence intervals and performance range for (a) GSM8K, (b) Math500, (c) Countdown, and (d) Sudoku.

## B.3. Compute-matched comparison with baselines

To evaluate training efficiency, we conducted a compute-matched experiment where we halved the epochs for GIFT and LIFT to directly compare them against Vanilla and CART at equivalent compute scales (e.g., 1 Vanilla epoch corresponds to 0.5 LIFT epochs). As shown in Table 10, while LIFT demonstrates performance comparable to baselines at lower epochs, it scales significantly better as training progresses. This sustained improvement stems from LIFT dynamically adjusting its training target difficulty based on the noise schedule. Ultimately, this confidence-based token selection acts as an effective adaptive curriculum (Parashar et al., 2025), systematically optimizing the learning process as the model improves.

*Table 10.* **Compute-matched comparison.** We compare GIFT and LIFT against Vanilla and CART at equivalent compute scales. LIFT scales favorably at higher compute budgets.

| Epochs (Vanilla/CART) | Epochs (GIFT/LIFT) | GSM8K | | | | MATH | | | |
|---|---|---|---|---|---|---|---|---|---|
| | | Vanilla | CART | GIFT | LIFT$_3$ | Vanilla | CART | GIFT | LIFT$_3$ |
| 1 | 0.5 | 76.6 | 78.8 | 76.3 | 76.8 | 33.6 | 35.2 | 35.0 | 34.0 |
| 2 | 1 | 79.6 | 78.9 | 76.9 | 75.8 | 33.8 | 37.0 | 34.0 | 34.6 |
| 4 | 2 | 77.1 | 78.4 | 78.5 | 78.9 | 33.8 | 34.8 | 34.6 | 35.9 |
| 8 | 4 | 77.4 | 76.4 | 76.6 | 77.7 | 32.6 | 32.0 | 33.8 | 35.7 |
| 16 | 8 | 77.3 | 76.8 | 77.8 | **80.2** | 32.2 | 31.4 | 31.4 | 36.1 |
| 20 | 10 | 77.3 | 76.5 | 77.7 | **80.2** | 32.6 | 29.2 | 31.8 | **36.6** |

## C. Dataset Construction Details for LIFT-SFT-12K

We constructed the dataset by mining and consolidating math-focused samples from three publicly available post-training corpora: NVIDIA/Llama-Nemotron-Post-Training-Dataset (Bercovich et al., 2025), open-r1/Mixture-of-Thoughts (Open-R1, 2025), and AllenAI/Dolci-Think-RL-32B (Team OLMo et al., 2025). From each source, we filtered instances specifically related to mathematical problem solving and reasoning tasks. The filtered subsets were then merged and randomly sampled to obtain a balanced collection of 12,000 examples. This curated dataset was used to fine-tune LLaDA-8B-Instruct.

*Table 11.* Hyperparameters used for training the model.

| Hyperparameter | Value |
|---|---|
| Learning rate scheduler type | Linear |
| Adam $\beta$ parameters | $\beta_1 = 0.9, \ \beta_2 = 0.999$ |
| Gradient accumulations steps | 4 |
| Per device train batch size | 2 |
| Epochs | 20 |
| Maximum sequence length | 4096 |
| Precision | bf16 |
| Lora $r$ | 128 |
| Lora $\alpha$ | 256 |
| Weight decay | 0.1 |
| Maximum gradient norm | 1.0 |

## D. Implementation Details

### D.1. Training

All methods are trained use the common hyperparameters listed in Table 11, with method-specific learning rates. For *vanilla*, we find that a learning rate of $1e-5$ yields the best performance. *CART* uses the same setting for consistency. For *GIFT*, we use the recommended learning rate of $2e-5$ on s1K, while a lower rate of $1e-6$ performs better on LIFT-SFT-12K. Across all settings, LIFT uses a learning rate of $5e-6$.

### D.2. Inference

**Evaluation Hyperparameters.** We follow the evaluation setup of the d1 (Zhao et al., 2025) for all experiments. The model generates 2 tokens per diffusion step and is evaluated with generation lengths of 128, 256, and 512 tokens. Decoding is performed with temperature $\tau = 0$.

For AIME, we use a temperature $\tau = 0.1$ for AIME'24 and $\tau = 0.2$ and AIME'25. The generation length is fixed to 512 and number of evaluation steps were 256. Additionally to speeden the evaluation, we implement prefix-caching (Wu et al., 2025).

# E. Additional Results on AIME'24 and AIME'25

*Table 12.* Performance comparison on AIME'24 and AIME'25 under different avg@$K$ and pass@$K$ values

| Method | AIME'24 | | | | AIME'25 | | | |
|---|---|---|---|---|---|---|---|---|
| | Avg8 | Pass8 | Avg16 | Pass16 | Avg8 | Pass8 | Avg16 | Pass16 |
| Instruct | 0.4 | 3.3 | 0.4 | 3.3 | 0.4 | 3.3 | 0.4 | 3.3 |
| Vanilla | 0.4 | 3.3 | 0.8 | 6.6 | 0.0 | 0.0 | 0.23 | 3.3 |
| GIFT | 1.3 | 6.7 | 2.1 | 16.7 | 0.0 | 0.0 | 0.0 | 0.0 |
| CART | 0.4 | 3.3 | 1.5 | 10.0 | 0.4 | 3.3 | 0.2 | 3.3 |
| **LIFT$_2$** | 0.8 | 6.7 | 1.1 | 10.0 | 0.0 | 0.0 | 0.4 | 6.7 |
| **LIFT$_3$** | 1.0 | 10.0 | 1.7 | 16.7 | 0.8 | 6.7 | 0.8 | 6.7 |

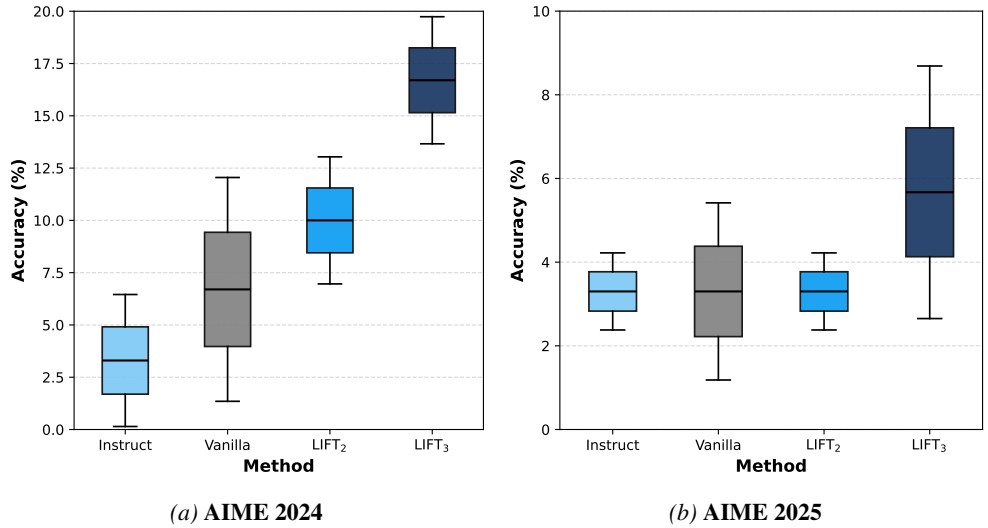

*(a)* **AIME 2024**          *(b)* **AIME 2025**

*Figure 8.* **Confidence Intervals for AIME 2024 and 2025.**

In Table 12, we provide expanded results on AIME'24 and AIME '25 on pass and average at k=8,16, with confidence intervals for AIME in Fig. 12.

# F. Additional Results on HumanEval and MBPP

We extend our evaluation to the domain of code generation, assessing model performance on MBPP (Austin et al., 2021b) and HumanEval (Chen et al., 2021). For this testing, models were first fine-tuned on the KodCode (Xu et al., 2025) dataset for 5 epochs and then evaluated with a maximum generation length of 256 tokens. As presented in Table 13, LIFT demonstrates strong performance on this task, achieving the best overall results compared to the baselines.

*Table 13.* **Evaluation on Code Generation.** Models were fine-tuned on the KodCode dataset for 5 epochs. We report performance on MBPP and HumanEval with a maximum generation length of 256 tokens. LIFT variants achieve the strongest overall results compared to existing baselines.

| Method | MBPP (256) | HumanEval (256) |
|---|---|---|
| Instruct (Base) | 41.1 | 34.8 |
| Vanilla | 43.2 | 31.1 |
| CART | 41.9 | 32.9 |
| GIFT | **44.4** | 35.2 |
| **LIFT$_2$** | 43.6 | **37.4** |
| **LIFT$_3$** | 44.0 | 36.3 |

