# OpenReview forum: "Learnability-Informed Fine-Tuning of Diffusion Language Models"
_ICML.cc/2026/Conference — ICML 2026 regular_

### Official Review · Reviewer_aHrw · 2026-02-13

**Soundness:** 3
**Presentation:** 3
**Significance:** 3
**Originality:** 3
**Overall Recommendation:** 5
**Confidence:** 4

**Summary:**

This paper studies supervised fine-tuning (SFT) for diffusion language models (DLMs) through the lens of token “learnability,” arguing that effective training depends jointly on what token is being predicted (proxied by token frequency) and when it is predicted along the diffusion time axis.

**Compliance With Llm Reviewing Policy:**

Affirmed.

**Key Questions For Authors:**

See weakness part.

**Limitations:**

See weakness part.

**Strengths And Weaknesses:**

The method story is somewhat shallow beyond the central heuristic. LIFT is a clean and reasonable design, but the core contribution is mainly a time-conditioned switch among Top-K / Bottom-K / random masking based on confidence (Sec. 5, Fig. 3), and the paper does not provide a deeper learning-theoretic or optimization-based justification for why this specific three-region schedule (controlled by H) should be close to optimal.
1. The choice of the partition parameter H is under-motivated and lightly explored. The text states that H=2 or H=3 “work well in practice,” and that larger H makes LIFT behave more like vanilla due to the middle region dominating (discussion around the “Approximate Variant of LIFT” / immediately above it, where the H behavior is described). This is plausible, but the paper does not give a clear criterion for selecting H or a broader sweep showing stability across tasks/models beyond a small set of values (Table 1 compares LIFT_2 and LIFT_3, but the selection rule remains informal).
2. The “compute efficiency” comparison is informative but not fully apples-to-apples across methods. Table 3 and Fig. 4 report H100 hours and include an RLVR oracle baseline (“d1 (oracle)”) with extremely large compute, while other baselines (vanilla, CART, GIFT, LIFT) sit in the 1–2 hour range. The message “LIFT is far cheaper than d1” is clear, but it would be helpful to separate “training method efficiency” from “overall pipeline cost,” since the oracle baseline is qualitatively different (Table 3 and the paragraph describing RLVR compute).
3. The approximate variant LIFT-A is described as a gating approximation, but the accuracy–cost trade-off is not unpacked in detail. The paper explains that LIFT-A avoids the second forward pass at t by computing loss at t+\rho and zeroing out unselected tokens via a gating mask (the “Approximate Variant of LIFT” paragraph). However, it is not fully clear when LIFT-A deviates from full LIFT in terms of training signal quality, and whether the approximation introduces systematic bias across diffusion times or token groups; the evidence is mostly aggregate benchmark numbers (Table 3, Fig. 4) rather than a targeted analysis.
4. The baseline coverage is good, but some comparisons would benefit from clearer alignment with the paper’s “what vs. when” framing. The text positions CART as focusing on when (re-weighting by local unmasked neighborhood) and GIFT as focusing on what (uncertainty-based masking at x_1), motivating LIFT as modeling the interaction (the “Baselines” paragraph in Sec. 6 setup). This framing is helpful, but the paper could strengthen the claim by reporting a direct diagnostic that the baselines indeed shift supervision toward the intended token groups/time regions (for example, a small analysis analogous to Fig. 2 but applied to the supervision distribution induced by CART/GIFT/LIFT).
5. Some evaluation details are deferred, and the main section could be more explicit on variance and statistical stability. The “Results” section mentions reporting mean over three runs (Sec. 6.2 opening), and the text notes confidence intervals are in the appendix (near Table 1 caption/nearby text). Given that some gains are large on AIME while smaller on other tasks, bringing a compact variance summary (or at least a consistent CI marker) into the main tables would make the empirical claims easier to audit.

Despite the limitations above, the paper has a clear problem statement, a simple and well-motivated method, and strong empirical results across multiple benchmarks, including convincing compute–performance evidence (Tables 1–3, Fig. 4). The main idea is good and is presented in a way that is easy to reuse in future work on DLM post-training. In my view, the paper is above the average level for ACL-style submissions, and I recommend accept.

---

> ### Author Rebuttal · Authors · 2026-03-31
>
> > Q1: On the justification and connection of LIFT to continuous diffusion
>
> Although LIFT might seem like a straightforward heuristic, its design is firmly rooted in continuous image diffusion literature, particularly concerning the shifts in optimization dynamics across varying Signal-to-Noise Ratios (SNR). High-noise regimes $ T \to 1$ inherently produce high-variance, uninformative gradients [1,2]. To address this in LIFT, we use Top-K masking to stabilize the MDLM ELBO by training only on the most learnable positions.
>
> Similarly, at low noise levels, continuous image diffusion models utilize re-weighting strategies to prevent wasting compute on easily predicted, high-SNR features [3]. LIFT instead utilizes Bottom-K masking to serve as a discrete form of hard example mining: by first passing inputs through at an increased noise level of t + rho, identifying the most appropriate targets to learn. While the discrete threshold parameter H is used to explicitly control these training boundaries, the underlying motivation for LIFT's design is rooted in the optimization phenomena observed in continuous diffusion. We will discuss this in the final version of our paper.
>
> > Q2: On the choice of parameter $H$ in LIFT
>
> Thank you for the suggestion. While Table 6 already ablates up to $H=4$, we have added a broader sweep for $H$ below. As $H$ increases, the middle region $(1/H, 1-1/H)$ expands. Because neither bottom-$k$ nor top-$k$ masking is applied in this interval, the LIFT objective naturally converges toward the vanilla MDLM loss.
>
> |  | GSM8K | M500 | CD | SU | AIME 24 | AIME 25 |
> | :---- | :---- | :---- | :---- | :---- | :---- | :---- |
> | Vanilla | 78.7 | 34.1 | 20.7 | 16.8 | 6.7 | 3.3 |
> | LIFT\_2 | 79.8 | 37.9 | 27.9 | 16.5 | 10.0 | 3.3 |
> | LIFT\_3 | 79.4 | 38.4 | 26.4 | 17.4 | 16.7 | 6.7 |
> | LIFT\_4 | 78.0 | 37.2 | 30.4 | 14.9 | 6.7  | 3.3  |
> | LIFT\_5 | 78.2 | 35.4 | 22.7 | 15.7 | 6.7 | 3.3 |
> | LIFT\_10 | 78.2 | 34.2 | 21.7 | 16.1 | 6.7 | 3.3 |
> | LIFT\_15 | 77.8 | 34.1 | 22.6 | 15.8 | 3.3 | 3.3 |
> | LIFT\_20 | 78.0 | 33.4 | 21.0 | 15.5 | 6.7 | 3.3 |
>
> > Q3: On the compute Matched Experiment for LIFT v/s baselines
>
> We conducted a compute-matched experiment where we halved the epochs for GIFT and LIFT to compare them directly against Vanilla and CART at equivalent compute scales (e.g., 1 Vanilla epoch vs. 0.5 LIFT epochs). The results are shown below, with results formatted as GSM8K / MATH500
>
> | Epochs (Vanilla/CART) | Epochs (GIFT/LIFT) | Vanilla | CART | GIFT | LIFT |
> | :---: | :---: | :---: | :---: | :---: | :---: |
> | **1** | **0.5** | 76.6 / 33.6 | 78.8 / 35.2 | 76.3 / 35.0 | 76.8 / 34.0 |
> | **2** | **1** | 79.6 / 33.8 | 78.9 / **37.0** | 76.9 / 34.0 | 75.8 / 34.6 |
> | **4** | **2** | 77.1 / 33.8 | 78.39 / 34.8 | 78.5 / 34.6 | 78.9 / 35.9 |
> | **8** | **4** | 77.4 / 32.6 | 76.4 / 32.0 | 76.6 / 33.8 | 77.7 / 35.7 |
> | **16** | **8** | 77.3 / 32.2 | 76.8 / 31.4 | 77.8 / 31.4 | **80.2** / 36.1 |
> | **20** | **10** | 77.3 / 32.6 | 76.5 / 29.2 | 77.7 / 31.8 | **80.2** / **36.6** |
>
> While LIFT has comparable performance to baselines at lower epochs, it scales better as training progresses. LIFT dynamically adjusts its training target difficulty based on the noise schedule, prioritizing relatively easier tokens during high-noise states ($T \to 1$) and targeting harder tokens at lower noise levels. Ultimately, this confidence-based token selection acts as an effective adaptive curriculum, systematically optimizing the learning process as the model improves.
>
> > Q4: Impact on ELBO by LIFT-A
>
> LIFT-A is a biased approximation of the MDLM ELBO (see response to **Reviewer LqPr (R2), Q2**). Evaluating the loss at $t+\rho$ shifts the expected masking rate to $\mathbb{E}[t+\rho] = 0.75$, biasing optimization towards states with higher noise. This creates a clear trade-off: standard LIFT provides an unbiased, MDLM objective, while LIFT-A trades exactness for compute efficiency.
>
> > Q5: On Token Supervision shift by GIFT/CART
>
> Thanks for the suggestion. We have conducted analyses on the scale of 3M tokens from s1k to demonstrate the supervision shift induced by GIFT and CART. We demonstrate that GIFT trains on rare tokens (*what*) and CART downweights token loss at higher masking rates (*when*).
>
> Figures:
> 1. https://anonymous.4open.science/r/icml2026_supp-B4AA/gift_frequency_analysis.pdf
> 2. https://anonymous.4open.science/r/icml2026_supp-B4AA/cart_weight_analysis.pdf
>
>
> > Q6: CI summary in the main text and for AIME.
>
> Thanks for the suggestion. We will add CIs as a subscript to each dataset in the main tables. CIs for AIME have also been included in our response to **Reviewer pKR8 (R3) - Q1**.
>
>
>
> [1] Campbell, A., et al. A Continuous Time Framework for Discrete Denoising Models. NeurIPS 2022
>
> [2] Kingma, D., et al. Variational Diffusion Models. NeurIPS 2021.
>
> [3] Choi, J.,et al. Perception Prioritized Training of Diffusion Models. CVPR 2022

---

> > ### Author Rebuttal · Reviewer_aHrw · 2026-04-01
> >
> > Thank you for your detailed rebuttal. However, I have already given a very positive score, so my final score will remain unchanged.

---

> > > ### Author Response · Authors · 2026-04-01
> > >
> > > Thank you for the insightful questions. We enjoyed engaging in the discussion during the review period
> > >
> > > Authors

---

### Official Review · Reviewer_pKR8 · 2026-02-22

**Soundness:** 3
**Presentation:** 3
**Significance:** 4
**Originality:** 2
**Overall Recommendation:** 5
**Confidence:** 5

**Summary:**

This paper proposes LIFT, a supervised fine-tuning strategy for masked diffusion language models (DLMs) that explicitly accounts for 1) what tokens are hard / easy based on model confidence, and 2) when they are learnable across diffusion time steps. LIFT performs time-dependent token selection: at high noise, it trains on top-K most confident tokens, at low noise, it trains on bottom-K least confident tokens; and uses vanilla masking strategies in the middle regime. Empirical experiments show gains over vanilla SFT over DLMs on several math benchmarks.

**Compliance With Llm Reviewing Policy:**

Affirmed.

**Final Justification:**

I recommend accept. The paper proposes a simple and practically useful fine-tuning strategy for masked diffusion language models, supported by a clear motivating analysis and solid empirical improvements. Overall, I found the work reasonably significant as a practical contribution to DLM post-training recipes.

My initial concerns were well addressed in the rebuttal through additional statistics, expanded evaluations, and robustness analysis, which materially increased my confidence in the method. After considering both the author response and the other reviews, I raised my score from 4 to 5.

**Key Questions For Authors:**

- Can you report variability across seeds or confidence intervals for AIME (and other high-variance benchmarks)?
- How does LIFT perform on non-math domains such as coding?
- Have you observed hacked behaviors induced by the confidence-based selection heuristic?

**Limitations:**

Not quite; see weaknesses and questions above.

**Strengths And Weaknesses:**

Strengths:
- LIFT is a simple technique to describe and implement upon existing supervised fine-tuning methods
- The direction of leveraging multiple forward passes of DLMs (or approximations as done in LIFT-A) to provide richer learning signals is well motivated and could be broadly useful for improving DLM (post-)training.

Weaknesses:
- Empirical results may be statistically fragile such as AIME; while the paper reports confidence intervals for some tasks, similar statistical reporting (more seeds and confidence intervals) on AIME would better support the central claim.
- The evaluation scope is concentrated on math tasks; it is unclear whether LIFT transfers to other important domains like coding.
- The confidence-based strategy changes the effective training distribution and may encourage some shortcut or hacked behaviors (e.g., focusing on particular token patterns or artifacts) relative to standard DLM masking. The paper would benefit from discussion and diagnostics of such potential biases.

---

> ### Author Rebuttal · Authors · 2026-03-31
>
> We thank Reviewer zZkj for their review of our paper, and address the main concerns below.
>
> > Q1: On confidence intervals for AIME
> Thanks for this suggestion. We have added the confidence intervals for AIME below.
>
> |  | AIME24 | AIME25 |
> | :---- | :---- | :---- |
> | Instruct | 3.3 (Mean) 1.61 (Std) | 3.3 (Mean) 0.47 (Std) |
> | Vanilla | 6.7 (Mean) 2.73 (Std) | 3.3 (Mean) 1.08 (Std) |
> | LIFT\_2 | 10.0 (Mean) 1.55 (Std) | 3.33 (Mean) 0.47 (Std) |
> | LIFT\_3 | 16.7 (Mean) 1.55 (Std) | 5.67 (Mean) 1.54 (Std) |
>
> We will include this in the revised version of our paper as well.
>
> > Q2: On the application of LIFT to coding.
>
> Thanks for the suggestion, we finetuned LLaDA for coding using LIFT on KodCode [1] (a SFT dataset for coding), and then evaluated on MBPP and HumanEval. All methods were trained for 5 epochs in total, and all the other hyperparameters were kept the same. For inference, we generated 256 tokens. As can be seen, LIFT performs well on this task, achieving the best results amongst baselines.
>
> |  | MBPP (256) | HumanEval (256) |
> | :---- | :---- | :---- |
> | Instruct | 41.1 | 34.8 |
> | Vanilla | 43.2 | 31.1 |
> | CART | 41.9 | 32.9 |
> | GIFT | 44.4 | 35.2 |
> | LIFT\_2 | 43.6 | 37.4 |
> | LIFT\_3 | 44.0  | 36.3 |
>
> > Q3: On the hacked behaviors induced due to confidence-based training of LIFT
>
> We appreciate the observation. While the confidence-based strategy does encourage the model to focus on specific token patterns, these learned patterns/behaviours are a feature designed to mitigate the "training-inference divide" of MDLMs.
>
> This divide occurs because MDLMs train on randomly-masked patterns, but rely on greedy unmasking during inference (prioritizing high-confidence positions). LIFT closes this gap by mimicking the greedy inference behavior, i.e., training on easy tokens at $T=1$ and hard tokens at $T=0$. The model needs to learn these specific patterns to align its training distribution with the actual generation trajectory.
>
> If these targeted behaviors introduced harmful biases, errors would compound rapidly. Instead, as shown in the table below, when generating multiple tokens per step (which strictly penalizes detrimental hacking), LIFT maintains higher accuracy and suffers a much smaller drop in performance. We will include this discussion in the final version of our paper.
>
> |  | GSM8K / tokens/step | MATH / tokens/step |
> | :---- | :---- | :---- |
> | Instruct (1/2/4) | 78.9 / 77.4 / 69.6 | 35.1 / 32.4 / 29.4 |
> | Vanilla (1/2/4) | 80.4 / 77.3 / 63.3  | 36.6 / 32.6 / 22.6 |
> | LIFT (1/2/4) | 81.8 / 80.2 / 70.5 | 37.8 / 36.8 / 31.2 |
>
> The figures for the table can be found at:
> 1. https://anonymous.4open.science/r/icml2026_supp-B4AA/gsm_tok_per_step.pdf
> 2. https://anonymous.4open.science/r/icml2026_supp-B4AA/math_tok_per_step.pdf
>
> [1] Xu, Z., et al. KodCode: A diverse, challenging, and verifiable synthetic dataset for coding. ACL, 2025.

---

> > ### Author Rebuttal · Reviewer_pKR8 · 2026-04-01
> >
> > I have read the rebuttal and thank the authors for the additional experiments and clarifications. The added AIME statistics, coding results, and robustness analysis address my main concerns well and increase my confidence in the method. While some uncertainty remains regarding broader generalization and possible heuristic biases, I believe the rebuttal materially strengthens the empirical support for the paper. Overall, my concerns are adequately addressed, and I will raise my score.

---

> > > ### Author Response · Authors · 2026-04-01
> > >
> > > Thank you for increasing the score. We are glad that you found our additional clarifications and experiments helpful.
> > >
> > > Authors

---

### Official Review · Reviewer_LqPr · 2026-03-10

**Soundness:** 2
**Presentation:** 2
**Significance:** 3
**Originality:** 3
**Overall Recommendation:** 4
**Confidence:** 3

**Summary:**

This paper studies supervised fine-tuning for diffusion language models. The authors observe that vanilla SFT ignores token learnability, which depends on both token difficulty and diffusion timestep. They propose LIFT, a learnability-informed training strategy that selects supervision targets based on model confidence and timestep. Experiments on math reasoning benchmarks show consistent improvements over existing SFT baselines and strong gains on AIME with substantially lower compute than RL methods.

**Compliance With Llm Reviewing Policy:**

Affirmed.

**Final Justification:**

The rebuttal adequately addresses my initial concerns. The new compute-matched baselines clarified the efficiency of the LIFT method, and the comparison experiment and analysis with LIFT-C explained the optimization's stability at boundaries. Additionally, the expanded ablations on sampling strategies and the inclusion of coding tasks empirically supported the method's robustness and generality. Since the main concern has been addressed, I recommend a weak accept.

**Key Questions For Authors:**

1. The method partitions the diffusion timestep into three regimes using fixed thresholds to determine the token selection strategy. While this design is intuitive, it would be helpful if the authors could provide additional theoretical intuition or empirical evidence supporting this choice. In particular, it would be valuable to understand whether the regime transitions could influence optimization behavior (e.g., gradient variance) near the boundaries.

2. In the approximate variant (LIFT-A), the loss is computed at timestep $t+\rho$ with a gating mask applied to selected tokens. It would be helpful if the authors could further clarify how this approximation relates to the original ELBO objective defined at timestep $t$. Additional discussion on the theoretical implications of this approximation would strengthen the presentation.

3. The secondary masking ratio $\rho$ is sampled uniformly to estimate token learnability. Has the potential impact of this stochastic sampling on the stability of the confidence estimation been investigated? It would be useful to know whether alternative strategies (e.g., fixed schedules, averaging over multiple samples, or variance-reduction techniques) were considered.

4. The experimental evaluation focuses primarily on mathematical and logical reasoning benchmarks. To better understand the generality of the proposed approach, it would be helpful if the authors could comment on whether LIFT has been evaluated, or is expected to perform similarly, on other tasks such as coding, commonsense reasoning, or general instruction-following benchmarks.

**Limitations:**

YES

**Strengths And Weaknesses:**

**Strengths**:

1. The authors provide a large-scale analysis of over 0.5 billion tokens to examine the relationship between token prediction difficulty (frequency) and available context (diffusion timesteps).

2. The Learnability-Informed Fine-Tuning method introduces a structured approach to target selection during training, adapting the masking strategy based on model confidence and diffusion time.

3. The method demonstrates measurable and consistent improvements over baseline SFT approaches (Vanilla, GIFT, CART) across standard reasoning benchmarks (GSM8K, MATH, Countdown, Sudoku).

**Weaknesses**:

1. While the paper emphasizes the efficiency of LIFT relative to RL-based approaches, the full LIFT training procedure requires two forward passes per step to estimate token confidence and compute the final loss. This effectively increases the computational cost compared to standard SFT. A more strictly compute-matched comparison with the Vanilla baseline would help clarify how much of the observed improvement stems from the training strategy itself rather than the additional compute.

2. The method partitions the continuous diffusion timestep into three discrete regimes with hard thresholds to determine the token selection strategy. Although this design is intuitive, the use of abrupt switching introduces discontinuities in the training objective. It would be helpful to provide further theoretical justification or empirical analysis to better understand the potential impact of these transitions on optimization stability.

3. In the efficient variant LIFT-A, the loss is computed at timestep $t+\rho$ with a gating mask that zeroes out unselected tokens. As a result, the optimization objective differs from the original formulation intended at timestep $t$. The paper would benefit from additional analysis or discussion clarifying the approximation introduced by this design and its potential implications for the underlying ELBO objective.

4. The secondary masking ratio $\rho$ is sampled uniformly for each training step to estimate token learnability. While this stochastic design is simple and flexible, relying on a single random draw may introduce variance in the confidence estimation used for token selection. Additional discussion or ablation studies on the stability of this estimation process could further strengthen the methodological justification.

5. Although the paper positions LIFT as a general supervised fine-tuning strategy for diffusion language models, the experimental evaluation is primarily limited to mathematical and logical reasoning benchmarks. Evaluating the approach on a broader range of tasks (e.g., general language understanding or generation) would help assess its generality and practical applicability.

---

> ### Author Rebuttal · Authors · 2026-03-31
>
> We thank Reviewer LqPr for their review and address their main concerns below.
>
> > Q: On the Compute Matched Comparison with baselines
>
> We conducted a compute-matched experiment by halving the epochs trained for GIFT and LIFT to compare them with Vanilla SFT and CART at equivalent compute scales (e.g., 1 Vanilla epoch vs. 0.5 LIFT epochs). While LIFT has comparable performance at lower epochs, it scales better as training progresses. For more details, please refer to our response to **Reviewer aHrw (R4) - Q3**.
>
> > Q: On the optimization dynamics of LIFT.
>
> To address this, we trained a continuous variant of LIFT (LIFT-C) where, instead of partitioning the diffusion time into three hard regimes, we gently decay each token selection strategy to ensure continuity (see Figure https://anonymous.4open.science/r/icml2026_supp-B4AA/lift_continuous.png). As shown below, the results are comparable:
>
> | Model | GSM8K | MATH500 |
> | :--- | :--- | :--- |
> | LIFT-C_3 | 79.5 | 35.4 |
> | LIFT_3 | 78.5 | 37.4 |
>
> While LIFT is discontinuous at the boundaries, a bigger challenge in diffusion optimization is the high variance caused by the range of Signal-to-Noise Ratios (SNR) encountered during training. Previous work [5] handled this by downweighting the loss as $T \to 1$. LIFT achieves this downweighting dynamically by using a Top-K policy to select tokens with the least loss (highest confidence). Conversely, we use Bottom-K to focus training on tokens with the highest loss (lowest confidence).
>
> Consequently, LIFT induces a higher variance in the loss within an $\epsilon$-ball at the boundaries; LIFT significantly reduces loss variance globally across training (Vanilla Loss STD: 0.54 vs. LIFT: 0.46, a 14.8% reduction). In summary, we deliberately sacrifice local continuity at the boundaries in exchange for lower global loss variance throughout the training process.
>
> > Q: On the relation between LIFT-A and the MDLM ELBO
>
> Thanks for this insightful question. For context, standard MDLM loss optimizes the continuous-time ELBO [1]. Since the masking rate $t \sim \mathcal{U}(0,1)$, the expected training masking rate is $\mathbb{E}[t] = 0.5$.
>
> **LIFT-A (Temporal Bias)**: To avoid an extra forward pass, LIFT-A evaluates the loss at a composite masking rate of $t+\rho$. Because $\rho \sim \mathcal{U}(0,1-t)$, the expected training masking rate shifts to $\mathbb{E}[t+\rho] = 0.75$. This biases the optimization objective toward states with higher noise.
>
> **LIFT (Modified Forward Process)**: Conversely, LIFT samples $t+\rho$ only to determine initial masked positions, but evaluates the loss at $t$ after downsampling the mask. It does not change the MDLM ELBO, but only alters the forward diffusion (masking) process. Altering the forward diffusion process has been explored in prior literature and has shown empirical benefits [2, 3].
>
> This creates a trade-off: standard LIFT provides an unbiased objective, while LIFT-A trades exactness for efficiency.
>
> > Q: On ablation of alternative sampling strategies for $\rho$
>
> We investigated alternative $\rho$ sampling strategies, including  fixed schedules, such as $\rho = \min(k, 1-t)$, and simple variance-reduced distributions, where $\rho \sim \mathcal{U}(0, \min(k, 1-t))$. In both cases, we experimented with $k \in \{0.1, 0.3\}$ to strictly bound $\rho$, and evaluate for max_gen_length=256 tokens.
>
> |  | GSM8K | MATH | Countdown | Sudoku |
> | :---- | :---- | :---- | :---- | :---- |
> | LIFT_3: $\\rho \~ U(0,1-t)$ | 79.4 | 37.6 | 26.4 | 7.9 |
> | Fixed: $rho \= min(0.1, 1-t) $ | 78.2 | 36.4 | 21.1 | 7.6 |
> | Fixed: $rho \= min(0.3, 1-t) $ | 77.9 | 34.6 | 21.1 | 8.5 |
> | Variance Reduced: $rho \= U(0.1, 1-t) $ | 78.9 | 34.4 | 20.0 | 4.5 |
> | Variance Reduced: $rho \= U(0.3, 1-t) $ | 78.4 | 36.2 | 22.8 | 6.5 |
>
> As shown, the highest-variance approach ($\rho \sim U(0,1-t)$) performs best. We attribute this to a strong regularization effect: a wide $\rho$ distribution maximizes the diversity of masking patterns that LIFT trains on. On the other hand, variance-reduced distributions limit and bias this masking diversity. For fine-tuning, this diversity acts as essential data augmentation, a dynamic similarly observed in image diffusion noise schedules [4]
>
> > Q: On experiments for Coding
>
> Thanks for the suggestion, we finetuned LLaDA for coding using LIFT. Due to space constraints, we have included this table in the response for **Reviewer pKR8 (R3) - Q2**. Results show that LIFT generalizes to coding, achieving the best results.
>
> [1] Sahoo, S. et al. Simple and effective masked diffusion language models. Neurips 2024
>
> [2] Li, S., et al. Lavida: A large diffusion model for vision-language understanding.  Neurips 2025
>
> [3] Xu, G., et al. Gift: Guided importance-aware fine-tuning for diffusion language models, 2025. arXiv:2509.20863.
>
> [4] Kingma, D., et al. Understanding diffusion objectives as the ELBO with simple data augmentation, Neurips 2023
>
> [5] Kingma, D., et al. Variational Diffusion Models. NeurIPS 2021.

---

> > ### Author Rebuttal · Reviewer_LqPr · 2026-04-01
> >
> > The authors addressed my concerns, and I appreciate their great efforts. I would like to raise the score from 3 to 4.

---

> > > ### Author Response · Authors · 2026-04-01
> > >
> > > Thank you for increasing the score. We appreciate the insightful discussions we had during our review phase.
> > >
> > > Authors

---

### Official Review · Reviewer_zZkj · 2026-03-16

**Soundness:** 3
**Presentation:** 3
**Significance:** 3
**Originality:** 3
**Overall Recommendation:** 5
**Confidence:** 3

**Summary:**

This paper develops a strategy for post-training DLMs. It identifies that SFT does not transfer cleanly to DLMs. To build the motivation, the authors conduct an empirical analysis of over 0.5 billion tokens across multiple post-training corpora and two DLM architectures (LLaDA and Dream), identifying a two-dimensional interaction between what tokens are learned (token frequency/difficulty) and at which diffusion timesteps they are learned. They find that rare tokens become effectively unlearnable at high noise levels, while common tokens offer little training signal at low noise levels. With this observation, the authors propose LIFT, a method that adaptively selects which tokens to supervise at each diffusion timestep. Easy/frequent tokens are selected at high noise, whereas hard/rare tokens at low noise, and standard SFT in between. The method is controlled by a single hyperparameter H. They also introduce a computationally cheaper single-pass variant, LIFT-A. Experiments on six mathematical reasoning benchmarks demonstrate consistent improvements over vanilla SFT and prior baselines (GIFT, CART), with notable gains on AIME'24 and AIME'25, while requiring much less compute than RL-based baselines.

**Compliance With Llm Reviewing Policy:**

Affirmed.

**Final Justification:**

The authors throughly adressed my concerns during the rebuttal period, and therefore, I wish to increase my score.

**Key Questions For Authors:**

Most of my questions are majorly covered under weaknesses, however to frame them formally and to add a few more for better clarity:
1. Why were LIFT training experiments not conducted on Dream? Does LIFT improve Dream's reasoning performance in the same way it improves LLaDA? If so, this would substantially strengthen the generality claim. If not, what might explain the discrepancy?
2.  LIFT uses model confidence pθ(x⁰ₖ|x_{t+ρ}) as an estimate of token difficulty. However, confidence under the current model during fine-tuning may change rapidly as the model is updated. How sensitive is LIFT to this dynamic? Is the confidence estimated from the model at the current training step, or from some earlier checkpoint?
3. The authors suggest LIFT "could potentially be useful for broader pre-training or instruction tuning settings." What would be the primary obstacles to applying LIFT during pre-training, where corpora are far larger and token frequency distributions may differ?

**Limitations:**

The authors don't discuss the limitations explicitly, however some of it is covered under the impact statement. Apart from that, it would be nice to include the following limitations:
1. The computational overhead of the two-pass LIFT mechanism.
2. Potential instability of confidence-based selection as the model updates during training.

**Strengths And Weaknesses:**

**Strengths**
1. The motivation is clear and the LIFT algo design is well motivated through the analysis.
2. Strong results: LIFT achieves up to a 3× relative improvement over vanilla SFT on AIME'24/AIME'25, and approaches RL-based methods at roughly 500–1000× lower compute cost.
3. The paper is well written and the related works section has convincingly addressed closest works GIFT and CART. The supplementary materials support reproducibility.

---
**Weaknesses**
1. LIFT requires two forward passes per training step, doubling the per-step compute relative to vanilla SFT. While LIFT-A reduces this, its performance is somewhat inconsistent (e.g., substantially worse on Sudoku than LIFT). A more detailed analysis of when LIFT-A is/ is not a reliable substitute would be valuable.
2. All main experiments use LLaDA-8B-Instruct and LLaDA-1.5. Dream is used only for the token-analysis plots, but not for LIFT training experiments. At least one main experiment on Dream would be useful for the generalization claim.

---

> ### Author Rebuttal · Authors · 2026-03-31
>
> We thank Reviewer zZkj for their review of our paper, and address the main concerns below.
>
> > Q1: On the extension of LIFT to DREAM
>
> Thank you for the suggestion. We have extended LIFT to DREAM. As shown below, LIFT transfers seamlessly and shows gains similar to those observed for LLaDA and LLaDA 1.5. We will include this table in the main paper.
>
> |  | GSM8K | MATH | Countdown | Sudoku |
> | :---- | :---- | :---- | :---- | :---- |
> | Instruct | 76.7 | 39.8 | 21.1 | 8.2 |
> | Vanilla | 76.1 | 30.6 | 25.0 | 14.8 |
> | GIFT | 78.5 | 40.0 | 23.4 | 16.0 |
> | CART | 77.8 | 38.9 | 22.3 | 17.2 |
> | LIFT\_2 | 77.9 | 40.8 | 33.6 | 22.5 |
> | LIFT\_3 | 79.1 | 40.6 | 25.6 | 17.5 |
>
> > Q2: On confidence-based token difficulty estimation and model stability during fine-tuning
>
> Thank you for raising these points. LIFT estimates token difficulty dynamically at each iteration using a time and memory-efficient no_grad forward pass, with no additional checkpoints. Importantly, our results show that LIFT is not sensitive to model updates during training. On the contrary, as discussed in Line 240, this confidence-based token selection acts as an effective adaptive curriculum, systematically optimizing the learning process to focus on tokens at the edge of model competence as the model improves.
>
> > Q3: On the application of LIFT beyond supervised post-training
>
> We appreciate the question. LIFT is best applied to pre-training once the model has attained a foundational level of competency, also known as the established 'mid-training' phase for pre-training autoregressive LLMs [1, 2, 3]. Because LIFT relies on a capable base model for reasonable confidence estimates, it is naturally suited for the mid-training stage and subsequent instruction fine-tuning. We will make this application of LIFT clearer in the revised manuscript.
>
> > Q4: On the inconsistent performance of LIFT-A on Sudoku
>
> We thank the reviewer for highlighting this compute-performance trade-off. While LIFT-A achieves strong results on Sudoku (Table 3, H=3), it is a biased approximation of the MDLM ELBO (see response to **Reviewer LqPr (R2) - Q2**). Evaluating the loss at $t+\rho$ shifts the expected masking rate to $\mathbb{E}[t+\rho] = 0.75$, biasing optimization toward higher noise regimes. This creates a clear trade-off: standard LIFT provides an unbiased, ideal objective, while LIFT-A trades exactness for compute efficiency. We will clarify this dynamic better in the main paper.
>
>
>
> [1] Dubey, A., Jauhri, A., Pandey, A., Kadian, A., Al-Dahle, A., Letman, A., ... & Meta GenAI. (2024). The Llama 3 Herd of Models. arXiv preprint arXiv:2407.21783.
>
> [2] NVIDIA. (2025). NVIDIA Nemotron 3: Efficient and Open Intelligence. arXiv preprint arXiv:2512.20856.
>
> [3] Team OLMo, Ettinger, A., Bertsch, A., Kuehl, B., Graham, D., Heineman, D., ... & Hajishirzi, H. (2025). OLMo 3. arXiv preprint arXiv:2512.13961.

---

> > ### Author Rebuttal · Reviewer_zZkj · 2026-04-03
> >
> > Thank you for kindly addressing my concerns and adding the experiments for DREAM. I have increased my score.

---

> > > ### Author Response · Authors · 2026-04-04
> > >
> > > Thank you for increasing the score. We appreciate the insightful discussions we had during our review phase.
> > >
> > > Authors

---

### Decision · Program_Chairs · 2026-04-30

**Decision:**

Accept (regular)

**Comment:**

The reviews were strongly positive and support acceptance. Reviewers found the paper well motivated, technically solid, and practically useful, with a clear empirical analysis and consistent gains over existing SFT baselines for diffusion language models. The rebuttal further strengthened the paper by addressing concerns about generalization, computation, coding transfer, and statistical stability, and several reviewers explicitly raised their scores afterward. Overall, the discussion suggests a simple but meaningful contribution with solid experimental support and clear relevance to DLM post-training.